# SARS-CoV-2 structural coverage map reveals viral protein assembly, mimicry, and hijacking mechanisms

Seán I O'Donoghue[1,2,3,*] iD, Andrea Schafferhans[1,4,5] iD, Neblina Sikta[1] iD, Christian Stolte[1] iD, Sandeep Kaur[1,3], Bosco K Ho[1], Stuart Anderson[2], James B Procter[6], Christian Dallago[5], Nicola Bordin[7] iD, Matt Adcock[2] & Burkhard Rost[5]

## Abstract

We modeled 3D structures of all SARS-CoV-2 proteins, generating 2,060 models that span 69% of the viral proteome and provide details not available elsewhere. We found that ~6% of the proteome mimicked human proteins, while ~7% was implicated in hijacking mechanisms that reverse post-translational modifications, block host translation, and disable host defenses; a further ~29% self-assembled into heteromeric states that provided insight into how the viral replication and translation complex forms. To make these 3D models more accessible, we devised a structural coverage map, a novel visualization method to show what is—and is not—known about the 3D structure of the viral proteome. We integrated the coverage map into an accompanying online resource (https://aquaria.ws/covid) that can be used to find and explore models corresponding to the 79 structural states identified in this work. The resulting Aquaria-COVID resource helps scientists use emerging structural data to understand the mechanisms underlying coronavirus infection and draws attention to the 31% of the viral proteome that remains structurally unknown or dark.

**Keywords** bioinformatics; COVID-19; data visualization; SARS-CoV-2; structural biology
**Subject Categories** Methods & Resources; Microbiology, Virology & Host Pathogen Interaction; Structural Biology
**Mol Syst Biol. (2021) 17: e10079**

## Introduction

Due to the COVID-19 pandemic, many life scientists have recently switched focus toward SARS-CoV-2 (severe acute respiratory syndrome coronavirus 2). This includes structural biologists, who have so far deposited > 1,000 entries in the Protein Data Bank (PDB; Berman et al, 2000) with details on the molecular conformation of the 27 viral proteins.

These structures are, in turn, driving molecular modeling studies, most focused on the spike glycoprotein (e.g., Jaimes et al, 2020; Gowthaman et al, 2021). Some modeling studies focus on breadth of coverage, predicting 3D structures for the entire SARS-CoV-2 proteome (Waman et al, 2021); this has been done using AlphaFold (preprint: Heo & Feig, 2020; Senior et al, 2020), C-I-TASSER (Zheng et al, 2021), MODELLER (Sedova et al, 2020; Srinivasan et al, 2020; Alsulami et al, 2021), Rosetta (preprint: Heo & Feig, 2020), and SWISS-MODEL (Waterhouse et al, 2018). Unfortunately, some of these methods have been found to give predictions that vary greatly (preprint: Heo & Feig, 2020), raising accuracy concerns; additionally, these approaches generally focus on deriving only one or a minimal number of structural states for each viral protein. To date, there has been no published, systematic analysis examining all structural states with supporting structural evidence.

Our goal in this work was to address these limitations via a depth-based strategy that models, for each viral protein, all states with related 3D structures in the PDB—this includes structures determined for other coronaviruses, such as SARS-CoV (severe acute respiratory syndrome coronavirus) or MERS-CoV (Middle East respiratory syndrome coronavirus), as well as many structures from more distantly related viruses, such as those causing polio or foot-and-mouth disease.

1  Garvan Institute of Medical Research, Darlinghurst, NSW, Australia
2  CSIRO Data61, Canberra, ACT, Australia
3  School of Biotechnology and Biomolecular Sciences (UNSW), Kensington, NSW, Australia
4  Department of Bioengineering Sciences, Weihenstephan-Tr. University of Applied Sciences, Freising, Germany
5  Department of Informatics, Bioinformatics & Computational Biology, Technical University of Munich, Munich, Germany
6  School of Life Sciences, The University of Dundee, Dundee, UK
7  Institute of Structural and Molecular Biology, University College London, London, UK
   *Corresponding author (lead contact). Tel: +61 2 9295 8329; E-mail: sean@odonoghuelab.org

Combining breadth and depth of coverage requires modeling methods with low computational cost; here, we use only sequence profile comparisons (Steinegger *et al*, 2019) to align SARS-CoV-2 sequences onto experimentally derived 3D structures (O'Donoghue *et al*, 2015). This generates what we call minimal models, in which 3D coordinates are not modified, but simply mapped onto SARS-CoV-2 sequences, with coloring used to indicate model quality (Heinrich *et al*, 2015).

Minimal models have substantial benefits: It is easy to understand how they were derived, helping assess the validity of insights gained. Thus, minimal models are broadly useful, even for researchers who are not modeling experts. Conversely, models generated by more sophisticated methods (e.g., Senior *et al*, 2020) can be more accurate, but it generally requires more time and expertise to assess their accuracy (e.g., preprint: Heo & Feig, 2020) and the validity of insights gained, thus limiting their usefulness.

Large numbers of models can be generated by such minimal strategies, raising a new problem: how to visually organize such complex datasets to be usable. This problem is one instance of what we consider to be the critical, central issue impeding not just COVID-19 research, but many areas of the life sciences (O'Donoghue *et al*, 2018): To address rapidly increasing data complexity, high-throughput machine learning studies of the kind presented in this work are not sufficient—the study outcomes also need to be accompanied with visual summarizes that help provide both insight and data navigation for other scientists (O'Donoghue, 2021). Thus, we introduce a novel concept: a one-stop visualization strategy that provides an overview of what is known—and not known—about the 3D structure of the viral proteome. This tailored visualization—called the SARS-CoV-2 structural coverage map—helps researchers find structural models related to specific research questions.

Once a structural model of interest is found, it can be used to explore the spatial arrangement of sequence features—i.e., residue-based annotations, such as nonsynonymous mutations or post-translational modifications. Here, we integrated the SARS-CoV-2 structural coverage map and 3D models into Aquaria (O'Donoghue *et al*, 2015), a web-based, molecular graphic system designed to simplify feature mapping and make minimal models broadly accessible to researchers who are not modeling experts. Previously, Aquaria could only map features from UniProt (The UniProt Consortium, 2019); for this work, we have added features from several additional sources, and we also refactored Aquaria to improve performance.

The resulting Aquaria-COVID resource (https://aquaria.ws/covid) comprises a large set of SARS-CoV-2 structural information not readily available elsewhere. The resource also identifies structurally dark regions of the proteome, i.e., regions with no significant sequence similarity to any protein region observed by experimental structure determination (Perdigão *et al*, 2015). Clearly identifying such regions helps direct future research to reveal viral protein functions that are currently unknown.

Below, we describe the resource and how the generated structural models provide new insights into the function of each viral protein, as well as general insights into how viral proteins self-assemble, and how they may mimic host proteins (Elde & Malik, 2009) and hijack host processes (Davey *et al*, 2011).

During the COVID-19 pandemic, the Aquaria-COVID resource aims to fulfill a vital role by helping scientists more rapidly explore and assess evidence for the molecular mechanisms that underlie coronavirus infection—and more easily keep abreast of emerging knowledge, as new 3D structures and sequence features become available.

## Results

Our study was based on 14 UniProt (The UniProt Consortium, 2019) sequences that comprise the SARS-CoV-2 proteome (Reagents and Tools Table). In PDB (Berman *et al*, 2000), these sequences were cross-referenced to 1,180 structures, all of which were determined for SARS-CoV-2. Using HHblits (Steinegger *et al*, 2019), we found an additional 880 related structures in PDB—initially determined for other organisms, but used here as minimal models for SARS-CoV-2 proteins. This gave a total of 2,060 matching structures (Datasets EV1–EV7) that collectively spanned 69% of the viral proteome (Dataset EV8), thus leaving the remaining 31% as structurally unknown or dark (Perdigão *et al*, 2015). The matching structures were incorporated into Aquaria (O'Donoghue *et al*, 2015), where they can be mapped with a wealth of features from UniProt, CATH (Dataset EV9; Sillitoe *et al*, 2021), SNAP2 (Hecht *et al*, 2015), and PredictProtein (Datasets EV10–EV12; Yachdav *et al*, 2014), in addition to user-defined features. These features include residue-based prediction scores for conservation, disorder, domains, flexibility, mutational propensity, subcellular location, and transmembrane helices (see Materials and Methods). To help other researchers use these models and features, we have extensively refactored Aquaria to improve cross-platform performance and created a matrix layout giving access to models for the 14 viral sequences (https://aquaria.ws/covid#matrix). We also created a structural coverage map

**Figure 1. SARS-CoV-2 structural coverage map.**

Integrated visual summary showing 79 distinct states found in 2,060 structural models derived by systematically comparing the SARS-CoV-2 proteome against all experimentally determined 3D structures. Viral proteins are shown as arrows scaled by sequence length, ordered by genomic location, and divided into three groups: (i) polyprotein 1a (top); (ii) polyprotein 1b (middle); and (iii) virion and accessory proteins (bottom). Above polyprotein 1a and 1b, a ruler indicates residue numbering from polyprotein 1ab; above selected accessory proteins, numbering indicates sequence length. Sequence regions with unknown structure are indicated with dark coloring. Regions that have matching structures are indicated with green coloring and with representative structures positioned below. Dark colored residues on the structure indicate amino acid substitutions, while conserved residues are colored to highlight secondary structure. Below the representative structures, graphs indicate three distinct states revealed in the matching structures: (i) viral protein hijacking of human proteins (gray coloring; Fig 3), (ii) human proteins that the viral protein may mimic (orange; Fig 2), or (iii) binding to antibodies, HLA, inhibitory peptides, RNA, or to other viral proteins (green; Fig 4). Bindings between viral proteins form two disjoint teams: (i) NSP7, NSP8, NSP9, NSP12, and NSP13 (parts of the viral replication and translation complex); and (ii) NSP10, NSP14, and NSP16. Nine viral proteins (called "suspects") had no structural evidence for interactions with other viral proteins, or for mimicry or hijacking of human proteins; seven of these (NSP2, NSP6, matrix glycoprotein, ORF6, ORF7b, ORF9c, and ORF10) are structurally dark proteins, i.e., have no significant similarity to any experimentally determined 3D structure. Representative structures for each state shown are given in Table 1; the complete list of matching structures is provided in Datasets EV1–EV3. Made using Aquaria and Keynote.

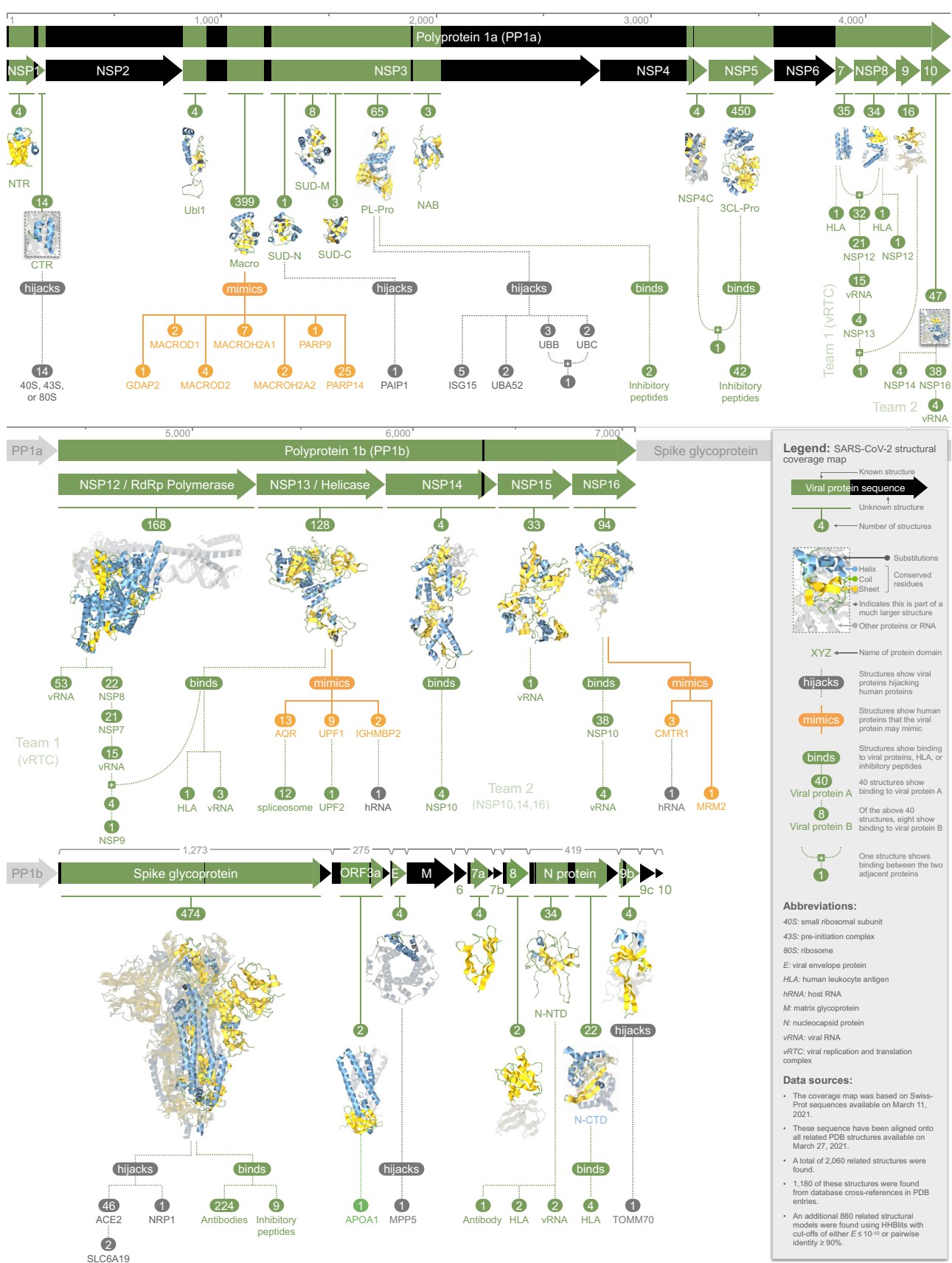

**Figure 1.**

(Fig 1)—a novel visual layout based on the viral genome organization. The coverage map summarizes key results obtained, including evidence for viral mimicry (Fig 2A and B) or hijacking of host proteins (Fig 3A–G), as well as viral protein interactions (Fig 4A and B). For each region with matching structures, an image of a single representative structure is shown in Fig 1, colored to convey alignment quality (Heinrich *et al*, 2015). Details about these structures are given in Table 1, which includes details on single, representative structures for each distinct state shown in the coverage map graphs (i.e., hijacking, mimicry, or binding). Each of these structural states can also be accessed via hyperlinks in the online version of the coverage map at the Aquaria-COVID resource (http://aquaria.ws/covid). In the following three sections of the Results, we systematically present key findings from structural models associated with three regions of the viral genome; these sections are intended to be read with close reference to Fig 1.

## Polyprotein 1a

Polyprotein 1a (a.k.a. PP1a) derives from polyprotein 1ab (a.k.a. PP1ab) and is cleaved into 10 proteins (NSP1–NSP10) that modify viral proteins, disable host defenses, and support viral replication.

NSP1 derives from residues 1–180 of PP1a and is thought to interact with the ribosome, suppressing translation of host mRNAs and promoting their degradation (Kamitani *et al*, 2009). We found four structures matching to an N-terminal region (NTR; PP1a 10–127), two of which were determined for SARS-CoV-2 and two from SARS-CoV. None of these structures showed either mimicry or interaction with other proteins or RNA; thus, on the coverage map (Fig 1), these structures are depicted using only a single representative image derived from one of the SARS-CoV-2 structures (Aquaria model https://aquaria.ws/P0DTC1/7k3n). Unusually, these structures provide few functional insights (Almeida *et al*, 2007), partly because they had a unique fold with no matches in CATH (Sillitoe *et al*, 2021). In contrast, a small C-terminal region (a.k.a. CTR; PP1a 145–180) had 14 matching structures, all derived from SARS-CoV-2 and all showing binding to various ribosome complexes, including 40S, 80S, and the 43S preinitiation complex (Fig 3A). These structures reveal that NSP1 obstructs host mRNA entry into the ribosome, thereby blocking innate immune responses (Thoms *et al*, 2020). On the coverage map, all structures, such as these, that show viral hijacking of human proteins are indicated via dark gray-colored graphs with dotted lines (Fig 1). NSP1 also had two short dark regions (Fig 1); the N-terminal dark region may be accounted for by high flexibility (average predicted B-value = 60; see Methods).

NSP2 (PP1a 181–818) may disrupt intracellular signaling by interacting with host proteins (Cornillez-Ty *et al*, 2009). Unfortunately, no structural information on these interactions is currently available, as NSP2 was found to be a dark protein, i.e., had no matching structures. This may be partly explained by the observation that, of all 15 PP1ab proteins, NSP2 had the highest predicted flexibility (average B-factor = 66), although it also had no predicted disorder (Dataset EV10).

NSP3 (PP1a 819–2,763) is a large, multidomain protein thought to perform many functions, including anchoring the viral replication complex to double-membrane vesicles derived from the endoplasmic reticulum (Lei *et al*, 2018). NSP3 had 483 matching structures—more than any other viral protein (Fig 1, Dataset EV7); these

structures clustered in 11 distinct sequence regions, each of which are described below.

NSP3 region 1 (a.k.a. Ubl1; PP1a 819–929) was the least conserved NSP3 region (average ConSurf score = 3.7; Dataset EV11), suggesting it adapts to host-specific defenses. Ubl1 is thought to bind single-stranded RNA and the viral nucleocapsid protein (Lei *et al*, 2018). Unfortunately, these interactions were absent in all four matching structures found (Dataset EV1), which all adopt a ubiquitin-like topology (CATH 3.10.20.350; Dataset EV9). Although it has distinct structural differences, Ubl1 may mimic host ubiquitin (Lei *et al*, 2018); however, we found no matches to structures of human ubiquitin, undermining the mimicry hypothesis.

NSP3 region 2 (PP1a 930–1,022) had no matches in CATH and no matching structures. This was the NSP3 region with lowest predicted sensitivity to mutation (median sensitivity 0%; Dataset EV11), highest predicted flexibility (average B-factor = 66), highest fraction of disordered residues (47%), and highest fraction of residues predicted to be solvent-accessible (99%). We speculate that this region acts as a flexible linker and may contain post-translational modification sites hijacking host signaling, as are often found in viral disordered regions (Davey *et al*, 2011).

NSP3 region 3 (PP1a 1,023–1,197) has a macro domain (CATH 3.40.220.10; Dataset EV9) that may counteract innate immunity via interfering with ADP-ribose (ADPr) modification (Lei *et al*, 2018). This was the second least conserved NSP3 region (ConSurf = 3.9; Dataset EV11) and had the highest fraction of mutationally sensitive residues (29%), suggesting it is well adapted to specific hosts. This region had 399 matching structures (Fig 1, Dataset EV1), of which 42 showed NSP3 aligned onto human proteins with moderately high significance ($E \sim 10^{-17}$), providing evidence for viral mimicry (Fig 2A, Dataset EV4). The potentially mimicked human proteins were as follows: GDAP2, MACROD1, MACROD2, MACROH2A1, MACROH2A2, PARP9, and PARP14—all of which are associated with ADPr modifications (Rack *et al*, 2016).

NSP3 region 4 (PP1a 1,198–1,230) was mostly disordered (1,210–1,230) and had no matching structures.

NSP3 region 5 (PP1a 1,231–1,353), also known as SUD-N, had one matching structure that adopted a macro-like topology (CATH 3.40.220.30). SUD-N is reported to bind RNA (Lei *et al*, 2018); however, the available structure shows this region from SARS-CoV hijacking PAIP1 (Fig 2B), a human protein implicated in translation initiation (Grosset *et al*, 2000).

NSP3 region 6 (PP1a 1,354–1,493), also known as SUD-M, has another macro-like domain (CATH 3.40.220.20) that may bind both RNA and host proteins, and take part in viral replication (Lei *et al*, 2018). However, these interactions were absent in all of the eight matching structures found (Dataset EV1). Comparing these with structures matching NSP3 region 3, we see considerable differences and no evidence of mimicry of host macro domains (Fig 1).

NSP3 region 7 (PP1a 1,494–1,562), also known as SUD-C, has a glutaredoxin-like topology (CATH 3.40.30.150) and had three matching structures, one of which also spanned NSP3 region 8. Based on analysis of these structures, Lei *et al* (2018) speculated that this region may bind metal ions and induce oxidative stress.

NSP3 region 8 (PP1a 1,563–1,881) comprises a papain-like protease (a.k.a. PL-Pro) thought to cleave NSP1–NSP3 from the polyprotein and to cleave ubiquitin-like modifications from host proteins (Fig 3C), thereby undermining interferon-induced antiviral

**A** NSP3 Mimicry of Host Proteins

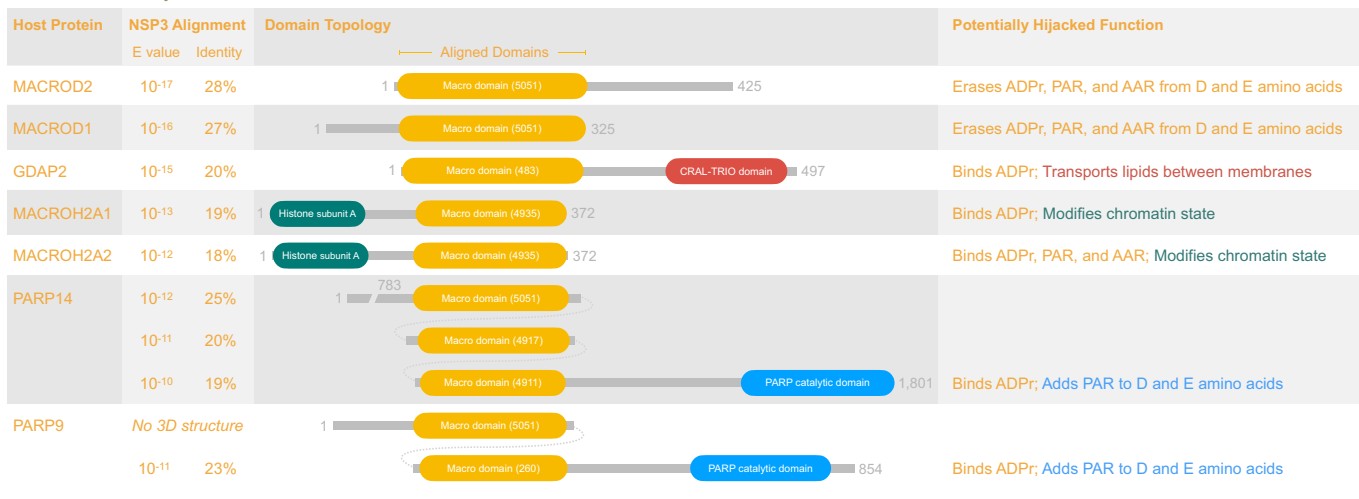

**B** NSP13 Mimicry of Host Proteins

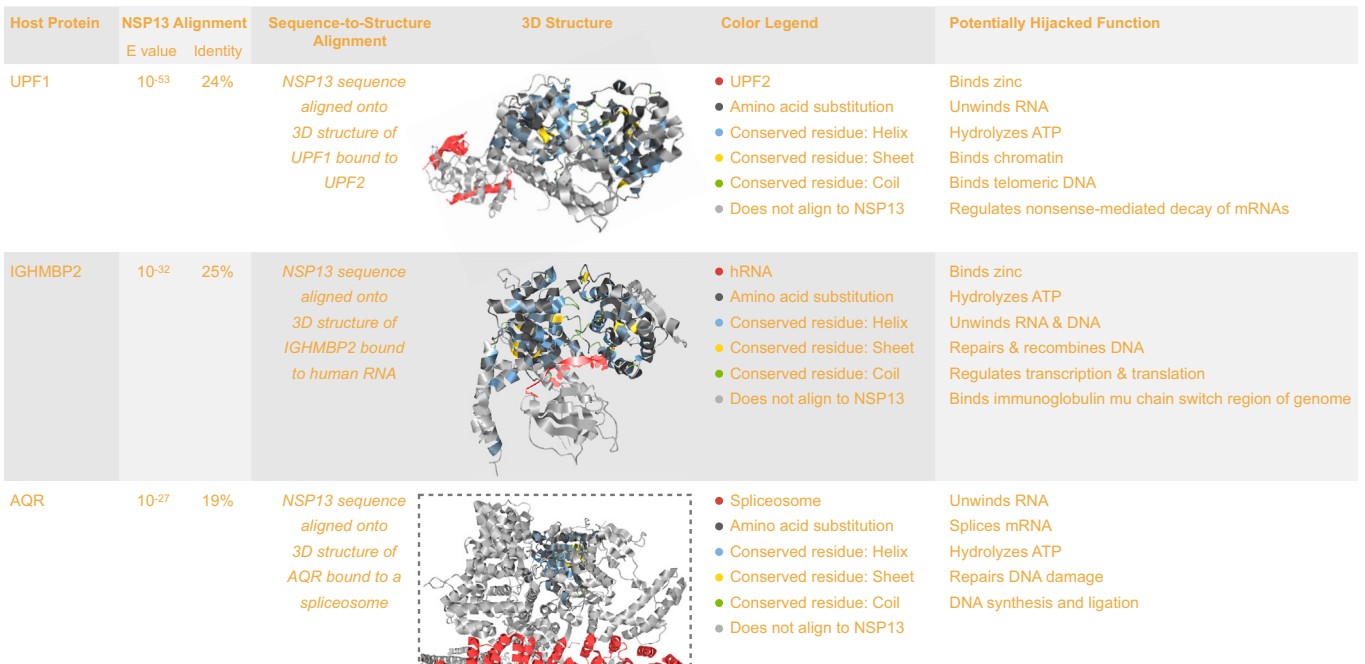

**Figure 2. Viral mimicry of human proteins.**

A Lists domain topology for seven human proteins potentially mimicked by the macro domain of NSP3. The list was ranked by alignment significance (HHblits E-value) and includes a summary of potentially mimicked functions. Each macro domain is numbered to indicate its CATH functional family. The top-ranked proteins (MACROD2 and MACROD1) remove ADPr from proteins, reversing the effect of ADPr writers (PARP14 and PARP9), and affecting ADPr readers (GDAP2, MACROH2A1, and MACROH2A2). For PARP9 and PARP14, the table indicates the best alignment of the NSP3 sequence onto the available structures corresponding to each macro domain.

B Lists three human helicase proteins potentially mimicked by NSP13. The list was ranked by alignment significance (HHblits E-value) and includes a summary of potentially mimicked functions. We found stronger evidence for mimicry by NSP13 than by NSP3. For each human protein, the 3D structure is shown with Aquaria's default coloring scheme, in this case indicating the region of alignment with NSP13 (Fig 1, Dataset EV4). For UPF1 (https://aquaria.ws/P0DTD1/2wjv), the structure coloring reveals that UPF2 binds to a region not matched by NSP13, suggesting that NSP13 may not bind UPF2. For IGHMBP2 (https://aquaria.ws/P0DTD1/4b3g), the structure coloring reveals that RNA binds to the region matched by NSP13, suggesting that NSP13 binds RNA. For AQR (https://aquaria.ws/P0DTD1/6jyt), the structure coloring reveals that the spliceosome binds to a region not matched by NSP13, suggesting that NSP13 may not bind the spliceosome.

Data information: Made using Aquaria, Photoshop, and Keynote.

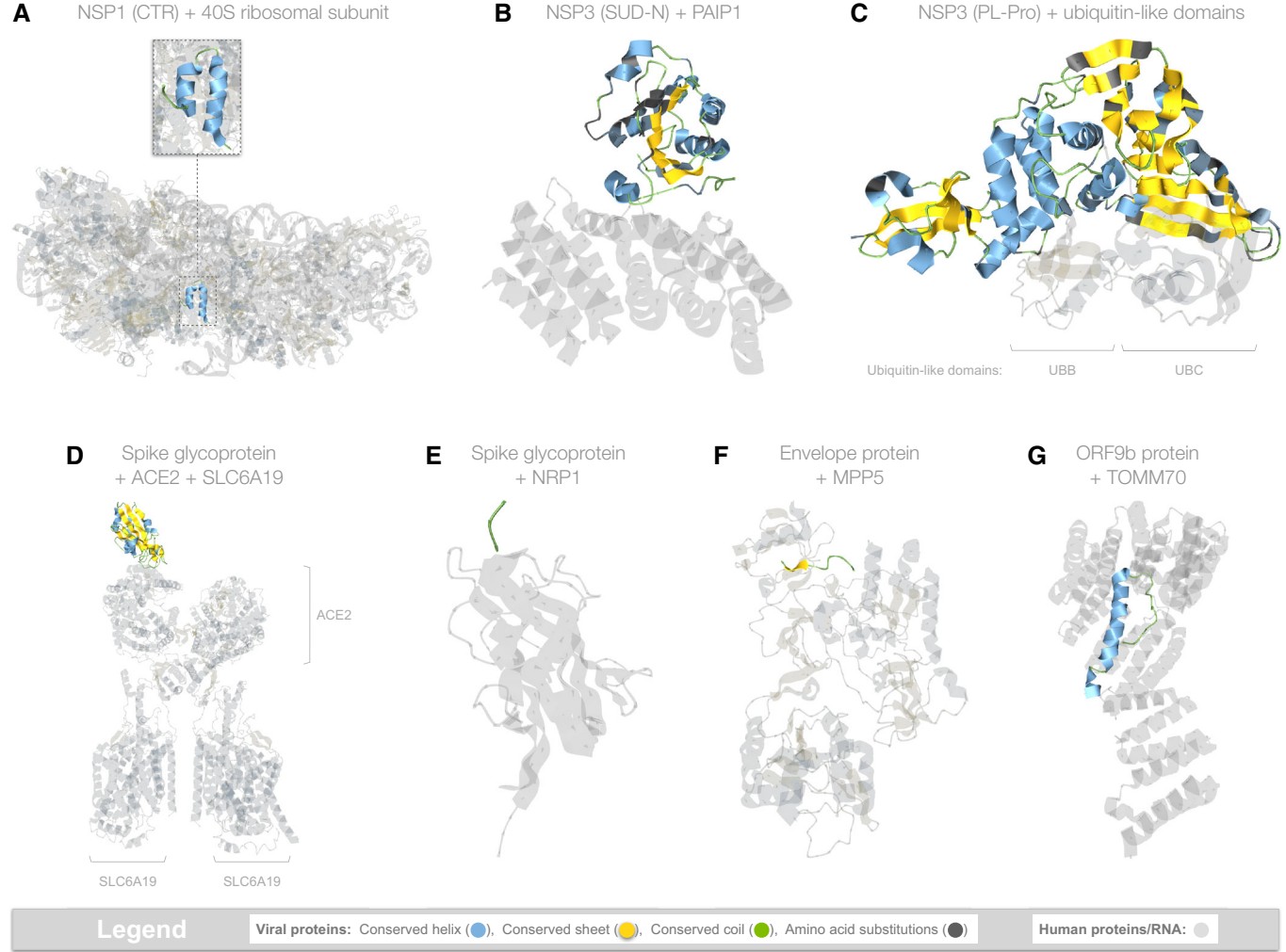

**A** NSP1 (CTR) + 40S ribosomal subunit

**B** NSP3 (SUD-N) + PAIP1

**C** NSP3 (PL-Pro) + ubiquitin-like domains

Ubiquitin-like domains: UBB | UBC

**D** Spike glycoprotein + ACE2 + SLC6A19

ACE2

SLC6A19 | SLC6A19

**E** Spike glycoprotein + NRP1

**F** Envelope protein + MPP5

**G** ORF9b protein + TOMM70

**Legend** — **Viral proteins:** Conserved helix (●), Conserved sheet (●), Conserved coil (●), Amino acid substitutions (●)    **Human proteins/RNA:** ●

**Figure 3. Viral hijacking of human proteins.**

Summarizes all structural evidence for viral hijacking; collectively, the regions shown cover 7% of the SARS-CoV-2 proteome. The structures are shown with Aquaria's default coloring scheme which, for viral proteins, highlights secondary structure as well as any amino acid substitutions from the SARS-CoV-2 sequence; human proteins and RNA are rendered as semi-transparent.

A  Hijacking of ribosomal complexes is shown in 14 matching structures, most of which were determined using the full-length sequence of NSP1 (180 residues); however, only a ~36 residue fragment was ordered enough to appear in the structures. The coloring scheme highlights the location of this fragment within the ribosome (https://aquaria.ws/P0DTC1/6zlw), revealing how NSP1 blocks host mRNA translation (Thoms *et al*, 2020).

B  Hijacking of PAIP1 (a.k.a. "PABP-interacting protein 1") is shown in only one matching structure that was determined using the SUD-N region of NSP3 from SARS-CoV (Nikulin *et al*, 2021). The structure (https://aquaria.ws/P0DTC1/6yxj) shows the strong overall sequence similarity in SARS-CoV-2 and reveals that, of the 15 residues contacting PAIP1, 13 are identical in SARS-CoV-2.

C  Hijacking of ubiquitin-like (Ubl) domains is shown in 10 matching structures, of which only one showed simultaneous binding to two Ubl domains (shown above). The structure (https://aquaria.ws/P0DTC1/5e6j) was determined using NSP3 from SARS-CoV (Békés *et al*, 2016), which had strong overall sequence similarity in SARS-CoV-2; of the 31 residues contacting UBB or UBC, 27 are identical in SARS-COV-2.

D  Hijacking of ACE2 is shown in 46 matching structures; however, only two also show binding to SLC6A19 (Yan *et al*, 2020). In the structure shown here (https://aquaria.ws/P0DTC2/6m17), spike glycoprotein does not directly bind to SLC6A19.

E  Hijacking of NRP1 (a.k.a. neuropilin-1) is shown in only one matching structure (https://aquaria.ws/P0DTC2/7jjc), which includes only a three-residue region from spike glycoprotein (Daly *et al*, 2020).

F  Hijacking of MPP5 (a.k.a. PALS1, "protein associated with Lin-7 1") is shown in only one matching structure (https://aquaria.ws/P0DTC4/7m4r), which includes only a nine-residue region from envelope protein (Liu & Chai, 2021).

G  Hijacking of TOMM70 (a.k.a. "translocase of outer mitochondrial membrane protein 70") is shown in only one matching structure (https://aquaria.ws/P0DTD2/7kdt), which includes only a 38-residue region from ORF9b protein (Gordon *et al*, 2020).

Data information: Made using Aquaria and Keynote.

activity (Lei *et al*, 2018). PL-Pro comprises three domains: one with ubiquitin-like topology (CATH 3.10.20.540), one with ruvA helicase-like topology (1.10.8.1190), and one with jelly roll topology (2.60.120.1680). PL-Pro had 65 matching structures, of which 12 showed binding to human ubiquitin-like proteins (Figs 1 and 3C): five showed binding to ISG15; two showed binding to UBA52; three

showed binding to UBB; and two showed binding to UBC. Of these 12 structures, one showed simultaneous binding to both UBB and UBC; this structure was determined for NSP3 from SARS-CoV; however, we expect that SARS-CoV-2 NSP3 is also likely to bind both UBB and UBC, based on strong sequence similarity (Fig 3C). We also expect that SARS-CoV-2 NSP3 is likely to bind UBA52, which has 100 and 99% sequence identity to UBB and UBC, respectively, in the regions shown in Fig 3C. Two additional matching structures showed the PL-Pro region in complex with inhibitory peptides.

NSP3 region 9 (PP1a 1,882–1,891) had no matching structures, but also no predicted disorder.

NSP3 region 10 (PP1a 1,892–2,021) comprises a nucleic-acid binding domain (a.k.a. NAB) thought to bind single-stranded RNA and to unwind double-stranded DNA (Lei *et al*, 2018). NAB had three matching structures that adopt a variant of the Rossmann fold unique to coronaviruses (CATH 3.40.50.11020).

NSP3 region 11 (PP1a 2,022–2,763) may anchor NSP3 to double-membrane vesicles (Lei *et al*, 2018). This was the most conserved NSP3 region (ConSurf = 4.9; Dataset EV11), suggesting it is less adapted to specific hosts. This region had no CATH matches and no matching structures.

NSP4 (PP1a 2,764–3,263) may act with NSP3 and NSP6 to create the double-membrane vesicles required for viral replication (Angelini *et al*, 2014). NSP4 mostly comprised a dark region (PP1a 2,764–3,167) with no CATH matches, no disorder, and multiple transmembrane helices. The C-terminal region (PP1a 3,168–3,263) comprised a domain called NSP4C with a DNA polymerase topology (CATH 1.10.150.420). This region matched to two structures from MHV-A59 (mouse hepatitis virus A59) and one from FCoV (feline coronavirus), all with a two-residue alignment gap at PP1a 3,197–3,198; these structures were all homodimers, yet NSP4C is thought to act primarily as a monomer (Xu *et al*, 2009). A final structure matched to only the last five residues of NSP4C and showed these residues in complex with NSP5 (Dataset EV1). The functional importance of this structure is unclear as it is not yet linked to a scientific publication—possibly, this structure shows a transient state associated with NSP4-NSP5 cleavage. Given these current uncertainties, we have not included this structure in Fig 4 or in the Discussion section on interaction between viral proteins.

NSP5 (a.k.a. 3CL-Pro; PP1a 3,264–3,569) is thought to cleave the viral polyprotein at 11 sites, resulting in NSP5–NSP16. NSP5 comprises two domains, one with thrombin-like topology (CATH 2.40.10.10) and another with a topology characteristic of viral proteases (1.10.1840.10). NSP5 had 450 matching structures, making it the third best characterized viral protein from a structural perspective (after spike glycoprotein; Fig 1, Dataset EV7). Many of these matching structures were determined to investigate methods for inhibiting the protease activity of NSP5; for example, 42 structures showed binding to inhibitory peptides.

NSP6 (PP1a 3,570–3,859) is a transmembrane protein thought to act with NSP3 and NSP4 to create double-membrane vesicles (Angelini *et al*, 2014). Like NSP2, NSP6 is a dark protein, with no matching structures—in addition, of the 15 PP1ab proteins, NSP6 is the least conserved (ConSurf = 4.4; Dataset EV10). NSP6 also had no CATH matches and no disordered regions.

NSP7 (PP1a 3,860–3,942) is a component of the viral replication and translation complex (a.k.a. RTC; te Velthuis *et al*, 2012); it had

35 matching structures, some showing interactions with other viral proteins (Figs 1 and 4A). In two structures, NSP7 occurred as a monomer, while 32 of the remaining structures showed NSP7 bound to NSP8. Of these 32 structures, 21 also showed binding to NSP12; of these, 15 structures also showed binding to viral RNA (a.k.a. vRNA); of these, four structures also showed binding to NSP13; of these, one structure also showed binding to NSP9. These structures provide insight into how RTC assembles (Fig 4A) and reveal that the NSP7 adopts a ruvA helicase-like topology (CATH 1.10.8.370), which comprises an antiparallel helical bundle with distinct substates, depending on its interaction partners. A final structure matched to only a nine-residue region of NSP7 and showed this region presented as an epitope via a complex with HLA (a.k.a. human leukocyte antigen).

NSP8 (PP1a 3,943–4,140) is another component of the replication and translation complex (te Velthuis *et al*, 2012). It features a highly conserved (ConSurf = 7.3) "tail" segment (PP1a 3,943–4,041), predominantly helical with some disordered residues and no CATH matches, followed by a less conserved (ConSurf = 5.7) "head" domain (PP1a 4,042–4,140) with alpha-beta plait topology (CATH 3.30.70.3540). NSP8 had 34 matching structures, all showing interactions to other proteins (Figs 1 and 4A). One structure showed binding to NSP12 only; this structure was determined for SARS-CoV NSP8; however, we inferred that NSP8 and NSP12 from SARS-CoV-2 may also interact, based on strong sequence similarity (Table 1 & Dataset EV6). Of the remaining matching structures, 32 showed binding to NSP7, with 21 also showing binding to NSP12; of these 21, 15 structures also showed binding to viral RNA; of these, four structures also showed binding to NSP13; of these, one structure also showed binding to NSP9. As noted for NSP7, these structures provide insight into how RTC assembles (Fig 4A). A final structure matched to only a nine-residue region of NSP8 and showed this region presented as an epitope via a complex with HLA.

NSP9 (PP1a 4,141–4,253) is believed to be another essential component of the RTC (Miknis *et al*, 2009). NSP9 had 16 matching structures with thrombin-like topology (CATH 2.40.10.250), most arranged in a homodimer, which is thought to be the functional state (Miknis *et al*, 2009). One of these structures showed binding to other RTC components (NSP7, NSP8, NSP12, NSP13, and viral RNA).

NSP10 (PP1a 4,254–4,392) is thought to act with NSP14 and NSP16 to cap and proofread RNA during genome replication (Bouvet *et al*, 2012). NSP10 had no CATH matches, yet had 47 matching structures, most showing interactions to other viral proteins (Figs 1 and 4B). In three matching structures, NSP10 was monomeric, while in two structures, NSP10 was a homododecamer that formed a hollow sphere (Fig 4B). Four matching structures showed binding to NSP14; these structures were determined for SARS-CoV proteins; however, we inferred that NSP8 and NSP12 from SARS-CoV-2 may also interact, based on strong sequence similarities (Table 1 & Dataset EV6). The remaining 38 structures showed binding to NSP16. Of these 38 structures, four also had viral RNA directly bound to NSP16, but not to NSP10.

**Polyprotein 1b**

Polyprotein 1b (a.k.a. PP1b) is cleaved by NSP5 into five proteins (NSP12–NSP16) that drive replication of viral RNA. These

proteins were predicted to have no disordered regions, no transmembrane helices, and to be conserved (ConSurf = 5.3–6.6; Dataset EV10).

NSP12 (PP1ab 4,393–5,324) is an RNA-directed RNA polymerase (a.k.a. RdRp) and is therefore the core component of the viral replication and translation complex (Yin *et al*, 2020). NSP12 was one of the more conserved PP1ab proteins (ConSurf = 6.5; Dataset EV10) and had a total of 168 matching structures. Of these, 53 showed

binding to viral RNA in the absence of other viral proteins (Fig 1); all of these 53 structures were determined for proteins from distantly related viruses (Table 1 & Dataset EV6); however, we inferred that SARS-CoV-2 NSP12 alone may also interact with viral RNA, based on SARS-CoV studies (te Velthuis *et al*, 2012). Of the remaining structures, 22 showed binding with NSP8; of these, 21 also showed binding with NSP7; of these, 15 also showed binding with viral RNA; of these, four also showed binding with NSP13; of

**A**  Viral interaction team 1 (NSP7, NSP8, NSP9, NSP12, NSP13)

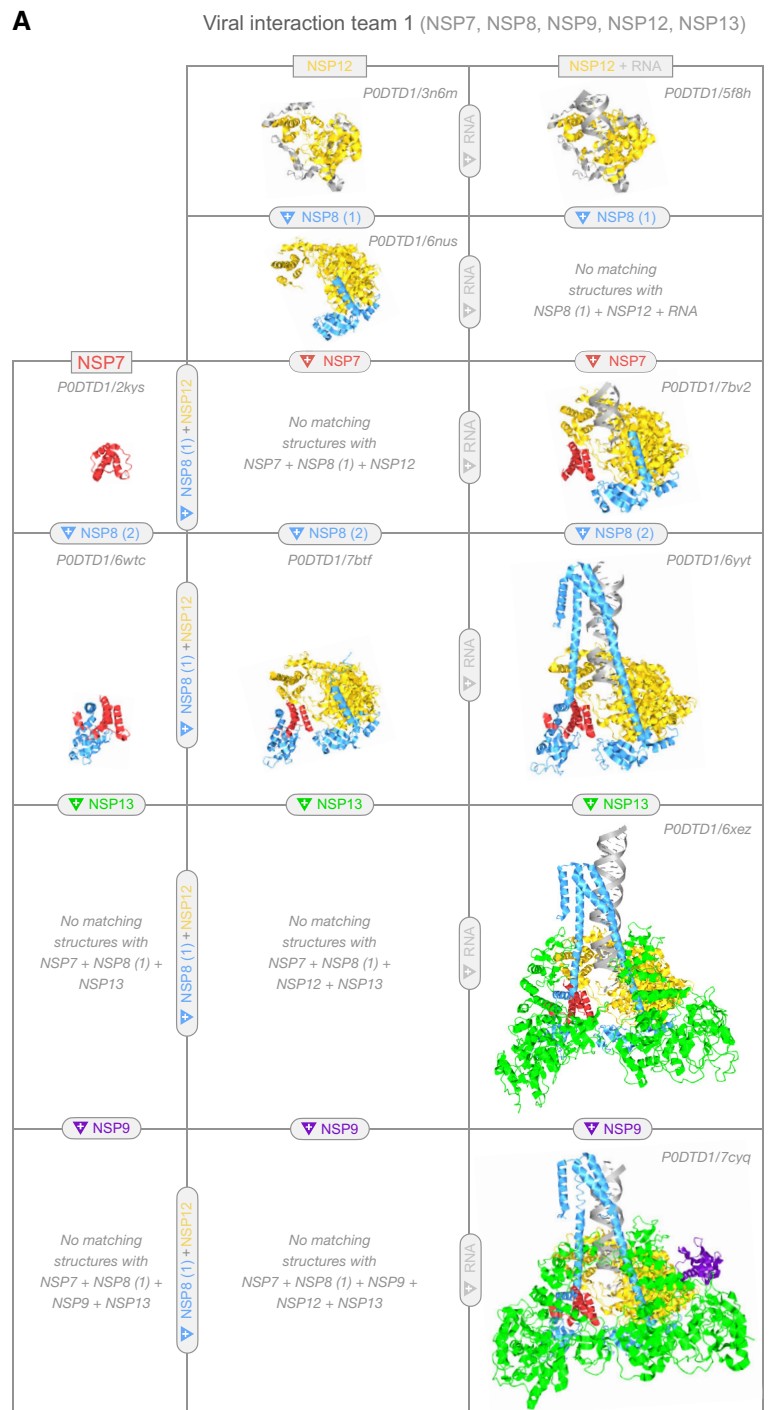

**B**  Viral interaction team 2 (NSP10, NSP14, NSP16)

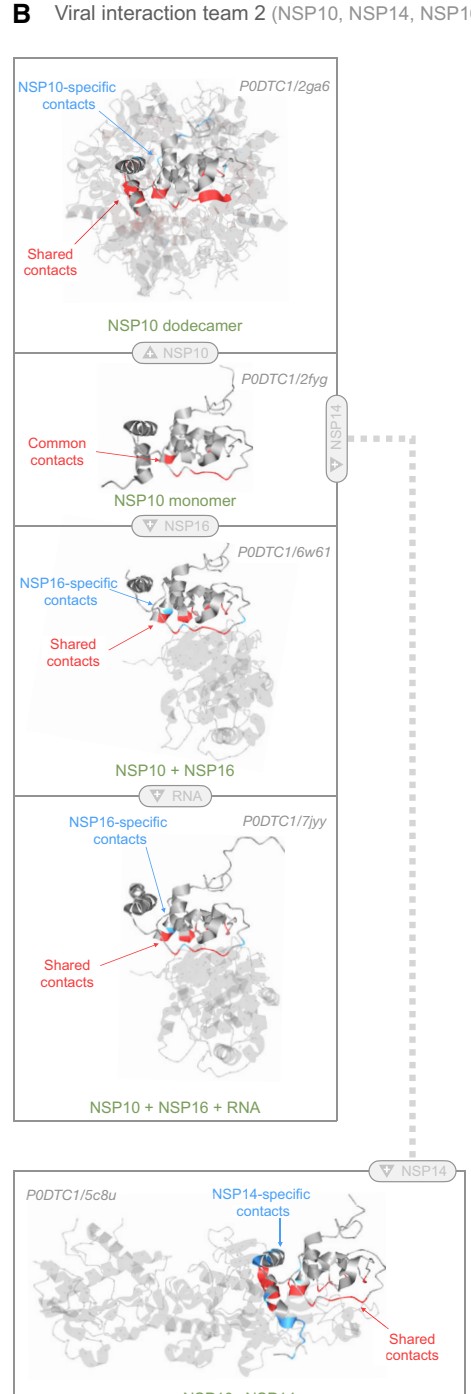

**Figure 4.**

◀

**Figure 4. Viral protein interaction teams.**

For each team, an assembly matrix is used to show all observed heteromeric states. For both teams, only a small subset of all combinatorially possible heteromeric states was observed; by highlighting possible transitions between observed states, the matrices suggest the order in which heteromers may assemble. Collectively, the regions shown cover 29% of the SARS-CoV-2 proteome.

A   In team 1, NSP7 (red), NSP8 (cyan), NSP9 (purple), NSP12 (yellow), and NSP13 (green) assemble into the replication and translation complex (RTC). NSP12 alone (top row, left) can replicate RNA (top row, right). NSP8 binds NSP12 at two sites: (i) at the NSP12 core (2[nd] row, left); and (ii) via NSP7-mediated cooperative interactions with NSP12 (4[th] row, center), greatly enhancing RNA replication (4[th] row, right). NSP7 + NSP8 alone form a dimer in most structures (4[th] row, left), but can also form a tetramer (e.g., https://aquaria.ws/P0DTD1/7jlt) or hexadecamer (e.g., https://aquaria.ws/P0DTD1/2ahm). Replication is also enhanced by NSP13 (5[th] row, right) and NSP9 (bottom row, right).

B   In team 2, NSP10 monomers (2[nd] row) can either self-assemble into a spherical dodecamer (top), dimerize with NSP14 (bottom row), or dimerize with NSP16 (third row). The NSP10 + NSP16 heterodimer was also seen bound to a three-residue RNA segment (fourth row). Residue coloring is used to show that NSP10, NSP14, and NSP16 appear to interact competitively, as noted in previous studies. In the structures shown, nine NSP10 residues (shown in red on the monomer) formed common intermolecular contacts in all three oligomers. Within each oligomer, most NSP10 residues involved in intermolecular contacts were shared (red) with at least one other oligomer; very few NSP10 residues formed contacts specific to that oligomer (blue).

Data information: For brevity, we omitted NSP9, NSP13, and NSP16 monomers, as well as the interaction between NSP4 and NSP5 (see Table 1). Made using Aquaria and Keynote.

these, one also showed binding with NSP9. As noted above, these structures provide insight into the order in which proteins assemble around NSP12 to form RTC (Fig 4A).

NSP13 (PP1ab 5,325–5,925) is a multi-functional helicase thought to play central roles in replication by unwinding double-stranded RNA (Subissi *et al*, 2014; Jang *et al*, 2020). The N-terminal half of NSP13 (PP1ab 5,325–5,577) had no matches in CATH, while the C-terminal half contained two Rossmann fold domains (CATH 3.40.50.300). NSP13 had 128 matching structures, of which only four showed binding to other RTC proteins (Fig 1, Dataset EV6). Three of the remaining structures showed direct binding to viral RNA; although these structures were determined for three very remote viruses (Table 1 & Dataset EV6), this interaction is consistent with *in vitro* SARS-CoV-2 studies (Jang *et al*, 2020). One of the remaining structures showed a nine-residue portion of NSP13 presented as an epitope by HLA, while 24 structures showed potential mimicry of three human helicase proteins (Fig 1). In nine of these 24 structures, the NSP13 Rossmann fold domains aligned onto human UPF1 with very high significance ($E \sim 10^{-53}$), providing evidence for viral mimicry. Of these nine structures, one showed UPF1 bound to UPF2; however, we concluded that there was insufficient evidence for direct UPF2 hijacking (Fig 2B); nonetheless, NSP13 mimicry may indirectly affect the UPF1/UPF2 interaction, so this is indicated in Fig 1 via green coloring. In two of the 24 structures, the NSP13 Rossmann fold domains aligned onto human IGHMBP2 with high significance ($E \sim 10^{-32}$), providing evidence for viral mimicry. Of these two, one structure showed IGHMBP2 bound to human RNA, suggesting that mimicry may lead to hijacking of human RNA (Fig 2B); this was indicated in Fig 1 via dark gray coloring. In 13 of the 24 structures, the first NSP13 Rossmann fold domain aligned onto human AQR with high significance ($E \sim 10^{-27}$), providing evidence for viral mimicry. Of these 13 structures, 12 showed AQR bound to the spliceosome; however, we concluded that there was insufficient evidence for direct spliceosome hijacking (Fig 2B); nonetheless, NSP13 mimicry may indirectly affect the AQR/spliceosome interaction, so this is indicated in Fig 1 via green coloring.

NSP14 (PP1ab 5,926–6,452) is a proofreading exoribonuclease thought to remove 3′-terminal nucleotides from RNA, thereby reducing mutations during viral genome replication (Minskaia *et al*, 2006). NSP14 had no matches in CATH, but had four matching structures, all in complex with NSP10 (Figs 1 and 4B).

NSP15 (PP1ab 6,453–6,798) is an uridylate-specific endoribonuclease thought to support viral genome replication (Ricagno *et al*,

2006). The N-terminal region of NSP15 had two domains—one with a double-stranded RNA-binding topology (CATH 3.30.160.820) and one with a Rossmann fold CATH 3.40.50.11580)—while the C-terminal region (PP1ab 6,642–6,798) had no matches in CATH. NSP15 had 33 matching structures, none of which showed potential mimicry or interactions with other proteins; however, one structure showed binding to viral RNA (Fig 1).

NSP16 (PP1ab 6,799–7,096) may methylate viral mRNA caps following replication, which is thought to be important for evading host immune defenses (Bouvet *et al*, 2010). This was also the most conserved PP1b protein (ConSurf = 6.6; Dataset EV10). NSP16 comprises a single Rossmann fold domain (CATH 3.40.50.150) that had a total of 94 matching structures. Of these structures, 38 showed binding to NSP10, of which four additionally showed binding to vRNA (Figs 1 and 4B). One additional matching structure showed NSP16 binding to vRNA in the absence of NSP10—however, this structure derived from a very remote virus (Dataset EV6) and had only marginal similarity ($E\text{-value} = 10^{-16}$); since this interaction was also inconsistent with SARS-CoV studies (Bouvet *et al*, 2010), we judged it to have insufficient evidence and did not include it in Figs 1 and 4, or Table 1. Another matching structure showed hijacking of human STAT2 by a distantly related flavivirus protein called NS5 (Dataset EV5), leading to suppression of host immune responses (Wang *et al*, 2020). However, the region of the NS5 structure in direct contact with STAT2 did not match the NSP16 sequence (Appendix Fig S1A); thus, we judged there was insufficient evidence to conclude that STAT2 hijacking occurs in SARS-CoV-2, and we did not include this potential hijacking on the coverage map (Fig 1, Table 1). Finally, four of the matching structures showed potential mimicry of the human RNA methyltransferase proteins CMTR1 and MRM2, with one of the matching structures also bound to human RNA (Fig 1). However, the alignments between these human proteins and NSP16 had only marginal $E$-values ($\sim 10^{-11}$; Dataset EV4); thus, while these proteins may share similar overall structure, it is not clear whether they share similar molecular function.

## Virion and accessory proteins

The remaining 3′ end of the genome encodes 12 proteins, many involved in virion assembly. Remarkably, our analysis found no interactions between these proteins.

Spike glycoprotein (a.k.a. S protein) binds host receptors, thereby initiating membrane fusion and viral entry (Hoffmann *et al*,

**Table 1.   SARS-CoV-2 minimal models used in Fig 1.**

| State[a] | 3D Model | Identity[b] | E[b] | Source[c] |
|---|---|---|---|---|
| NSP1 (NTR) | https://aquaria.ws/P0DTC1/7k3n | 100% | – | SARS-CoV-2 (Semper *et al*, 2021) |
| NSP1 (CTR) hijacks 40S, 43S, and 80S | https://aquaria.ws/P0DTC1/6zlw | 100% | – | SARS-CoV-2 (Thoms *et al*, 2020) |
| NSP3 (Ubl1) | https://aquaria.ws/P0DTC1/2gri | 77% | $10^{-21}$ | SARS-CoV (Serrano *et al*, 2007) |
| NSP3 (macro) | https://aquaria.ws/P0DTC1/6woj | 100% | – | SARS-CoV-2 (https://doi.org/10.2210/pdb 6WOJ/pdb) |
| NSP3 (macro) mimics GDAP2 | https://aquaria.ws/P0DTC1/4uml | 20% | $10^{-15}$ | Human (https://doi.org/10.2210/pdb4UML/pdb) |
| NSP3 (macro) mimics MACROD1 | https://aquaria.ws/P0DTC1/2x47 | 27% | $10^{-160}$ | Human (Chen *et al*, 2011) |
| NSP3 (macro) mimics MACROD2 | https://aquaria.ws/P0DTC1/4iqy | 28% | $10^{-16}$ | Human (Jankevicius *et al*, 2013) |
| NSP3 (macro) mimics MACROH2A1 | https://aquaria.ws/P0DTC1/1zr5 | 19% | $10^{-13}$ | Human (Kustatscher *et al*, 2005) |
| NSP3 (macro) mimics MACROH2A2 | https://aquaria.ws/P0DTC1/2xd7 | 18% | $10^{-12}$ | Human (https://doi.org/10.2210/pdb2XD7/pdb) |
| NSP3 (macro) mimics PARP9 | https://aquaria.ws/P0DTC1/5ail | 23% | $10^{-10}$ | Human (https://doi.org/10.2210/pdb5AIL/pdb) |
| NSP3 (macro) mimics PARP14 | https://aquaria.ws/P0DTC1/3q6z | 29% | $10^{-12}$ | Human (Forst *et al*, 2013) |
| NSP3 (SUD-N) + PAIP1 | https://aquaria.ws/P0DTC1/6yxj | 69% | $10^{-21}$ | SARS-CoV (https://doi.org/10.2210/pdb6XYJ/pdb) |
| NSP3 (SUD-M) | https://aquaria.ws/P0DTC1/2jzd | 80% | $10^{-23}$ | SARS-CoV (Chatterjee *et al*, 2009) |
| NSP3 (SUD-C) | https://aquaria.ws/P0DTC1/2kqw | 78% | $10^{-34}$ | SARS-CoV (Johnson *et al*, 2010a) |
| NSP3 (PL-Pro) | https://aquaria.ws/P0DTC1/6wrh | 100% | – | SARS-CoV-2 (https://doi.org/10.2210/pdb 6WRH/pdb) |
| NSP3 (PL-Pro) hijacks ISG15 | https://aquaria.ws/P0DTC1/6xa9 | 100% | – | SARS-CoV-2 (Klemm *et al*, 2020) |
| NSP3 (PL-Pro) hijacks UBA52 | https://aquaria.ws/P0DTC1/4rf0 | 31% | $10^{-31}$ | MERS-CoV (Bailey-Elkin *et al*, 2014) |
| NSP3 (PL-Pro) hijacks UBB | https://aquaria.ws/P0DTC1/4wur | 30% | $10^{-30}$ | MERS-CoV (Lei & Hilgenfeld, 2016) |
| NSP3 (PL-Pro) hijacks UBC | https://aquaria.ws/P0DTC1/4mm3 | 83% | $10^{-30}$ | SARS-CoV (Ratia *et al*, 2014) |
| NSP3 (PL-Pro) hijacks UBB + UBC | https://aquaria.ws/P0DTC1/5e6j | 82% | $10^{-30}$ | SARS-CoV (Békés *et al*, 2016) |
| NSP3 (PL-Pro) binds inhibitory peptides | https://aquaria.ws/P0DTC1/6wuu | 99% | – | SARS-CoV-2 (Rut *et al*, 2020) |
| NSP3 (NAB) | https://aquaria.ws/P0DTC1/2k87 | 82% | $10^{-19}$ | SARS-CoV (Serrano *et al*, 2009) |
| NSP4 | https://aquaria.ws/P0DTC1/3vcb | 59% | $10^{-37}$ | MHV-A59 (Xu *et al*, 2009) |
| NSP4 binds NSP5 | https://aquaria.ws/P0DTC1/7kvg/C | 99% | – | SARS-CoV-2 (https://doi.org/10.2210/pdb 7KVG/pdb) |
| NSP5 (3CL-Pro) | https://aquaria.ws/P0DTC1/5rfa | 100% | – | SARS-CoV-2 (https://doi.org/10.2210/pdb 5RFA/pdb) |
| NSP5 binds inhibitory peptides | https://aquaria.ws/P0DTC1/7bqy | 100% | – | SARS-CoV-2 (Jin *et al*, 2020) |
| NSP7 | https://aquaria.ws/P0DTC1/2kys | 98% | $10^{-33}$ | SARS-CoV (Johnson *et al*, 2010b) |
| NSP7 binds HLA | https://aquaria.ws/P0DTC1/7lg3 | 100% | – | SARS-CoV-2 (https://doi.org/10.2210/pdb 7LG3/pdb) |
| NSP7 binds NSP8 | https://aquaria.ws/P0DTC1/6m5i/A | 100% | – | SARS-CoV-2 (https://doi.org/10.2210/pdb 6M5I/pdb) |
| NSP7 binds NSP8 + NSP12 | https://aquaria.ws/P0DTC1/6m71/C | 100% | – | SARS-CoV-2 (Gao *et al*, 2020) |
| NSP7 binds NSP8 + NSP12 + vRNA | https://aquaria.ws/P0DTC1/7aap/C | 100% | – | SARS-CoV-2 (Naydenova *et al*, 2021) |
| NSP7 binds NSP8 + NSP12 + vRNA + NSP13 | https://aquaria.ws/P0DTC1/6xez/C | 95% | – | SARS-CoV-2 (Chen *et al*, 2020) |
| NSP7 binds NSP8 + NSP12 + vRNA + NSP13 + NSP9 | https://aquaria.ws/P0DTC1/7cyq/C | 100% | – | SARS-CoV-2 (Yan *et al*, 2021a) |
| NSP8 | https://aquaria.ws/P0DTC1/6m5i/B | 100% | – | SARS-CoV-2 (https://doi.org/10.2210/pdb 6M5I/pdb) |
| NSP8 binds NSP12 | https://aquaria.ws/P0DTC1/6nus/B | 97% | $10^{-76}$ | SARS-CoV (Kirchdoerfer & Ward, 2019) |
| NSP8 binds HLA | https://aquaria.ws/P0DTC1/7lg2 | 100% | – | SARS-CoV-2 (https://doi.org/10.2210/pdb 7LG2/pdb) |

**Table 1** (continued)

| State[a] | 3D Model | Identity[b] | E[b] | Source[c] |
|---|---|---|---|---|
| NSP9 | https://aquaria.ws/P0DTC1/6wxd | 98% | – | SARS-CoV-2 (https://doi.org/10.2210/pdb 6WXD/pdb) |
| NSP10 | https://aquaria.ws/P0DTC1/2g9t | 96% | $10^{-72}$ | SARS-CoV (Su et al, 2006) |
| NSP10 binds NSP14 | https://aquaria.ws/P0DTC1/5c8u/A | 95% | $10^{-68}$ | SARS-CoV (Ma et al, 2015) |
| NSP10 binds NSP16 | https://aquaria.ws/P0DTC1/6w61/B | 99% | – | SARS-CoV-2 (https://doi.org/10.2210/pdb 6W61/pdb) |
| NSP12 | https://aquaria.ws/P0DTD1/6yyt | 100% | – | SARS-CoV-2 (Hillen et al, 2020) |
| NSP12 binds vRNA | https://aquaria.ws/P0DTD1/3koa | 15% | $10^{-14}$ | FMDV (Ferrer-Orta et al, 2010) |
| NSP13 | https://aquaria.ws/P0DTD1/6jyt | 100% | $10^{-63}$ | SARS-CoV (Jia et al, 2019) |
| NSP13 mimics AQR | https://aquaria.ws/P0DTD1/4pj3 | 20% | $10^{-27}$ | Human (De et al, 2015) |
| NSP13 mimics AQR + spliceosome | https://aquaria.ws/P0DTD1/6id0 | 20% | $10^{-27}$ | Human (Zhang et al, 2019) |
| NSP13 mimics UPF1 | https://aquaria.ws/P0DTD1/2wjy | 24% | $10^{-53}$ | Human (Clerici et al, 2009) |
| NSP13 mimics UPF1 + UPF2 | https://aquaria.ws/P0DTD1/2wjv | 24% | $10^{-53}$ | Human (Clerici et al, 2009) |
| NSP13 mimics IGHMBP2 | https://aquaria.ws/P0DTD1/4b3f | 25% | $10^{-32}$ | Human (Lim et al, 2012) |
| NSP13 mimics IGHMBP2 + hRNA | https://aquaria.ws/P0DTD1/4b3g | 26% | $10^{-31}$ | Human (Lim et al, 2012) |
| NSP13 binds vRNA | https://aquaria.ws/P0DTD1/4n0o | 21% | $10^{-19}$ | Arterivirus (Deng et al, 2014) |
| NSP13 binds HLA | https://aquaria.ws/P0DTD1/7lfz | 100% | – | SARS-CoV-2 (https://doi.org/10.2210/pdb 7LFZ/pdb) |
| NSP14 | https://aquaria.ws/P0DTD1/5nfy | 95% | $10^{-142}$ | SARS-CoV (Ferron et al, 2018) |
| NSP15 | https://aquaria.ws/P0DTD1/6wxc | 97% | – | SARS-CoV-2 (Kim et al, 2021) |
| NSP15 binds vRNA | https://aquaria.ws/P0DTD1/6x1b | 97% | – | SARS-CoV-2 (Kim et al, 2021) |
| NSP16 | https://aquaria.ws/P0DTD1/6w4h | 99% | – | SARS-CoV-2 (Rosas-Lemus et al, 2020) |
| NSP16 mimics CMTR1 | https://aquaria.ws/P0DTD1/4n49 | 14% | $10^{-11}$ | Human (Smietanski et al, 2014) |
| NSP16 mimics MRM2 | https://aquaria.ws/P0DTD1/2nyu | 22% | $10^{-11}$ | Human (https://doi.org/10.2210/pdb2NYU/pdb) |
| NSP16 mimics CMTR1 + hRNA | https://aquaria.ws/P0DTD1/4n48 | 14% | $10^{-11}$ | Human (Smietanski et al, 2014 |
| NSP16 binds vRNA + NSP10 | https://aquaria.ws/P0DTD1/7jyy/A | 100% | – | SARS-CoV-2 (https://doi.org/10.2210/pdb7JYY/pdb) |
| Spike glycoprotein | https://aquaria.ws/P0DTC2/6vxx | 97% | – | SARS-CoV-2 (Walls et al, 2020) |
| Spike glycoprotein hijacks ACE2 | https://aquaria.ws/P0DTC2/7ct5 | 100% | – | SARS-CoV-2 (Guo et al, 2021) |
| Spike glycoprotein hijacks ACE2 + SLC6A19 | https://aquaria.ws/P0DTC2/6m17 | 100% | – | SARS-CoV-2 (Yan et al, 2020) |
| Spike glycoprotein hijacks NRP1 | https://aquaria.ws/P0DTC2/7jjc | 100% | – | SARS-CoV-2 (Daly et al, 2020) |
| Spike glycoprotein binds antibodies | https://aquaria.ws/P0DTC2/6w41 | 100% | – | SARS-CoV-2 (Yuan et al, 2020a) |
| Spike glycoprotein binds inhibitory peptides | https://aquaria.ws/P0DTC2/5zvm | 88% | $10^{-33}$ | SARS-CoV (Xia et al, 2019) |
| ORF3a | https://aquaria.ws/P0DTC3/6xdc | 100% | – | SARS-CoV-2 (preprint: Kern et al, 2020) |
| ORF3a binds APOA1 | https://aquaria.ws/P0DTC3/7kjr | 100% | – | SARS-CoV-2 (preprint: Kern et al, 2020) |
| Envelope protein | https://aquaria.ws/P0DTC4/5x29 | 85% | $10^{-35}$ | SARS-CoV (Surya et al, 2018) |
| Envelope protein hijacks MPP5 | https://aquaria.ws/P0DTC4/7m4r | 100% | – | SARS-CoV-2 (https://doi.org/10.2210/pdb 7M4R/pdb) |
| ORF7a | https://aquaria.ws/P0DTC7/6w37 | 100% | – | SARS-CoV-2 (https://doi.org/10.2210/pdb 6W37/pdb) |
| ORF8 | https://aquaria.ws/P0DTC8/7jtl | 99% | – | SARS-CoV-2 (Flower et al, 2021) |
| Nucleocapsid protein (NTD) | https://aquaria.ws/P0DTC9/6yi3 | 96% | – | SARS-CoV-2 (https://doi.org/10.2210/pdb6YI3/pdb) |
| Nucleocapsid protein (NTD) binds antibody | https://aquaria.ws/P0DTC9/7cr5 | 100% | – | SARS-CoV-2 (Daly et al, 2020) |
| Nucleocapsid protein (NTD) binds HLA | https://aquaria.ws/P0DTC9/7kgr | 100% | – | SARS-CoV-2 (Szeto et al, 2021) |
| Nucleocapsid protein (NTD) binds vRNA | https://aquaria.ws/P0DTC9/7acs | 96% | – | SARS-CoV-2 (Dinesh et al, 2020) |

**Table 1** (continued)

| State[a] | 3D Model | Identity[b] | E[b] | Source[c] |
|---|---|---|---|---|
| Nucleocapsid protein (CTD) | https://aquaria.ws/P0DTC9/6yun | 98% | – | SARS-CoV-2 (Zinzula et al, 2021) |
| Nucleocapsid protein (CTD) binds HLA | https://aquaria.ws/P0DTC9/7kgo | 100% | – | SARS-CoV-2 (Szeto et al, 2021) |
| ORF9b | https://aquaria.ws/P0DTD2/6z4u | 100% | – | SARS-CoV-2 (https://doi.org/10.2210/pdb 6Z4U/pdb) |
| ORF9b hijacks TOMM7 | https://aquaria.ws/P0DTD2/7kdt | 100% | – | SARS-CoV-2 (Gordon et al, 2020) |

[a]This table lists 79 distinct protein structural states found in this work, each with details on one representative minimal model, indicated using an Aquaria identifier. The indicated models correspond to those used to generate representative images and hyperlinks in the online version of Fig 1.
[b]In cases showing potential mimicry, the identity scores and E-values indicate similarity between the SARS-CoV-2 viral protein and a human protein.
[c]Indicates the organism used to derive the corresponding PDB structure as well as the publication associated with the PDB entry; where no publication is yet available, the DOI for the dataset is given. Organism names are abbreviated as follows: FMDV (foot-and-mouth disease virus); MERS-CoV (Middle East respiratory syndrome coronavirus); MHV-A59 (mouse hepatitis virus A59); SARS-CoV (severe acute respiratory syndrome coronavirus); SARS-CoV-2 (severe acute respiratory syndrome coronavirus 2).

2020). This protein has five domains: one with jelly roll topology (CATH 2.60.120.960), two alpha-beta plaits (3.30.70.1840), and two heptad repeat regions, called HR1 (1.20.5.300) and HR2 (1.20.5.790). Aquaria found a total of 474 matching structures—making this the second best characterized viral protein from a structural perspective (Fig 1, Dataset EV7). Of the matching structures, 46 showed hijacking of ACE2 (a receptor for SARS-CoV and SARS-CoV-2 entry); in two of these structures, ACE2 was also bound to SLC6A19, which had no direct contact with spike glycoprotein (Fig 3D). An additional matching structure showed hijacking of NRP1, another receptor that may facilitate viral entry (Fig 3E). Two further matching structures showed binding between DPP4 and spike glycoprotein from MERS-CoV—however, based on analysis of these structures (Appendix Fig S1B) combined with previous in vitro studies (Discussion), we considered that the evidence does not support binding between DPP4 and spike glycoprotein from SARS-CoV-2. Thus, potential hijacking of DPP4 was not included in Fig 1, although the matching structure evidence is included in Dataset EV5. Finally, 224 matching structures showed spike glycoprotein bound to antibodies, and nine structures showed binding to inhibitory peptides.

ORF3a may act as a homotetramer, forming an ion channel in host cell membranes that helps with virion release (Lu et al, 2006). ORF3a had two matching structures, both showing the protein as a dimer and thought to represent inactivated states; one structure additionally showed binding to human APOA1, which was used as an experimental technique to study ORF3a in lipid nanodiscs (preprint: Kern et al, 2020). Thus, APOA1 binding does not represent hijacking, and this interaction is therefore indicated on the coverage map with green coloring (Fig 1).

The envelope protein (a.k.a. E protein) matched four structures, of which one was a monomer and two showed a homopentamer, thought to span the viral envelope and form an ion channel (Surya et al, 2018). Finally, one structure showed a nine-residue region of E protein hijacking MPP5 (a.k.a. PALS1), a human protein normally associated with intercellular junctions (Fig 3F).

The matrix glycoprotein (a.k.a. M protein) is also thought to be part of the viral envelope (Vennema et al, 1996). The matrix glycoprotein had no matching structures.

ORF6 may block expression of interferon-stimulated genes (e.g., ISG15) that have antiviral activity (Frieman et al, 2007). ORF6 had no matching structures.

ORF7a may interfere with the host cell surface protein BST2, preventing it from tethering virions (Taylor et al, 2015). ORF7a had four matching structures that adopt an immunoglobulin-like topology (CATH 2.60.40.1550); this fold is believed to facilitate hijacking of monocyte interactions, thereby dysregulating immune responses (Zhou et al, 2021).

ORF7b is an integral membrane protein thought to localize to the Golgi compartment and the virion envelope (Schaecher et al, 2007). ORF7b had no matching structures.

ORF8 is thought to inhibit type 1 interferon signaling (Li et al, 2020); it is also very different to proteins from other coronaviruses. ORF8 had two matching structures, both showing a disulfide-linked homodimer assembly; these structures had a similar fold to ORF7a, but had dimerization interfaces not seen in other coronaviral proteins—these interfaces may allow ORF8 to form large-scale assemblies that mediate immune suppression and evasion (Flower et al, 2021).

The nucleocapsid protein (a.k.a. N protein) is thought to package the viral genome during virion assembly through interaction with the matrix glycoprotein, and also to become ADP-ribosylated (Grunewald et al, 2018). Depending on its phosphorylation state, this protein may also switch function, translocating to the nucleus and interacting with the host genome (Surjit et al, 2005). This protein had 56 matching structures clustered in two distinct regions. The N-terminal region (a.k.a. N-NTR; 28–30, 35–184) had 34 matching structures, of which two were homotetramers, eight were homodimers, 14 were monomers, two showed binding to viral RNA, and one showed binding to an antibody. In addition, two structures matched to nine-residue regions of N protein, showing these regions presented as epitopes by HLA (Szeto et al, 2021), while a final structure showed a six-residue region assembled as an homo-16-mer that is implicated in amyloid formation (preprint: Tayeb-Fligelman et al, 2021). The C-terminal region (a.k.a. N-CTR; 217–230, 243–365) had 22 matching structures, of which 16 were homodimers, four showed nine-residue regions of N protein presented as epitopes by HLA (Szeto et al, 2021), and two showed six-residue regions assembled as homo-16-mers that are implicated in amyloid formation (preprint: Tayeb-Fligelman et al, 2021). These structures suggest the oligomerization and RNA-binding activities of SARS-CoV-2 N protein may be disrupted by therapeutic strategies based on small molecule inhibitors developed for HCoV-OC43 (human coronavirus OC43) and MERS-CoV (Peng et al, 2020).

ORF9b is a lipid-binding protein thought to interact with mitochondrial proteins, thereby suppressing interferon-driven innate immune responses (Shi *et al*, 2014). ORF9b matched four structures, of which three show a homodimer assembly bound to a lipid analog (Meier *et al*, 2006). In the remaining structure, ORF9b adopts a very different 3D conformation and occurs as part of a heterodimer. This structure shows hijacking of TOMM70 (Fig 3G), a protein of the outer mitochondrial membrane that plays an important role in interferon response.

ORF9c (a.k.a. ORF14) is currently uncharacterized experimentally; it is predicted to have a single-pass transmembrane helix. ORF9c had no matching structures.

ORF10 is a predicted protein that currently has limited evidence of translation (Gordon *et al*, 2020), has no reported similarity to other coronavirus proteins, and has no matching structures.

# Discussion

The 2,060 matching structures found in this study capture essentially all SARS-CoV-2 protein states with direct, supporting structural evidence. We used these states to create a structural coverage map (Fig 1): a concise yet comprehensive visual summary of what is known—and not known—about the 3D structure of the viral proteome. Remarkably, we found so few states showing viral self-assembly (Fig 4), mimicry (Fig 2), or hijacking (Fig 3) that—excluding non-human host proteins—all states could be easily included in the coverage map by adding several rather simple graphs. This may indicate that host interactions are rarely used in COVID-19 infection, consistent with the notion that viral activity is largely shielded from the host. However, other experimental techniques have found many more interactions between viral proteins (Pan *et al*, 2008), and with host proteins (Gordon *et al*, 2020). Thus, the small number of interactions found in this work more likely indicates limitations in currently available structural data. We note that it may be possible to infer many more states from the available structural data via a range of focused methods (Smith & Sternberg, 2002; Franzosa & Xia, 2011; Kaján *et al*, 2014; Du *et al*, 2017; Gervasoni *et al*, 2020).

Based on a systematic, semi-automated analysis of the 2,060 matching structures, we could divide the 27 SARS-CoV-2 proteins into four categories: mimics, hijackers, teams, and suspects (Fig 1, Dataset EV7)—below, we highlight key insights derived within each of these categories.

## Mimics

In this work, we use the term mimic to describe viral proteins that are similar to host proteins in structure and function (Elde & Malik, 2009). We found structural evidence of mimicry for ~6% of the viral proteome (Datasets EV1 and EV2), comprising only three SARS-CoV-2 proteins: NSP3, NSP13, and NSP16 (Dataset EV4).

NSP3 may mimic host proteins containing macro domains, thereby hijacking ADP-ribose (ADPr) modifications and suppressing host innate immunity (Lei *et al*, 2018). We found seven potentially mimicked proteins (Fig 2A); the top-ranked matches (MACROD2 and MACROD1) remove ADPr from proteins

(O'Sullivan *et al*, 2019), thus reversing the effect of ADPr writers (e.g., PARP9 and PARP14, found in lymphoid tissues), and affecting ADPr readers (e.g., the core histone proteins MACROH2A1, and MACROH2A2, found in most cells). Thus, we speculate that, in infected cells, ADPr erasure by NSP3 may hijack epigenetic regulation of chromatin state (Schäfer & Baric, 2017), potentially contributing to variation in COVID-19 patient outcomes. Furthermore, in infected macrophages, activation by PARP9 and PARP14 may be hijacked by NSP3's erasure of ADPr, potentially contributing to the vascular disorders (Iwata *et al*, 2016) seen in COVID-19 (Varga *et al*, 2020).

NSP13 may mimic three human helicases, based on stronger alignment evidence than for NSP3 mimicry (Fig 2). However, we found no evidence for mimicry of the ~100 other human helicases (Umate *et al*, 2011), suggesting that NSP13 mimicry may hijack specific functions performed by the three helicases. The strongest evidence was for mimicry of UPF1 (a.k.a. regulator of nonsense transcripts 1, or RENT1), which acts in the cytoplasm as part of the nonsense-mediated mRNA decay pathway, known to counteract coronavirus infection (Wada *et al*, 2018); we speculate that UPF1 mimicry may hijack this pathway, thus impeding host defenses. UPF1 also acts in the nucleus, interacting with telomeres; we speculate that UPF1 mimicry may be implicated in the connection seen between COVID-19 severity, age, and telomere length (Aviv, 2020). The next strongest evidence was for mimicry of IGHMBP2 (a.k.a. immunoglobulin μ-binding protein 2, or SMBP2), which acts in the cytoplasm as well as the nucleus, where it interacts with single-stranded DNA in the class switching region of the genome (Yu *et al*, 2011), close to *IGMH*, the gene coding the constant region of immunoglobulin heavy chains. We speculate that IGHMBP2 mimicry may be implicated in the dysregulation of immunoglobulin-class switching observed clinically (Bauer, 2020). If these speculations about IGHMBP2 or UPF1 mimicry are correct, they suggest that NSP13 may sometimes switch roles, from viral replication to undermining host immunity via host genome interactions.

Finally, our analysis suggested that NSP16 may mimic the RNA methyltransferase proteins CMTR1 and MRM2. Since CMTR1 is implicated in interferon response (Haline-Vaz *et al*, 2008), this mimicry could be a mechanism used by SARS-CoV-2 to undermine host immunity. However, by the alignment criteria used in this work (O'Donoghue *et al*, 2015), we considered the evidence for NSP16 mimicry to be marginal and much weaker than the evidence for NSP3 and NSP13 mimicry (Dataset EV4). Thus, we judged that a more detailed analysis of NSP16 mimicry—as presented in Fig 2 for NSP3 and NSP13—was not warranted.

## Hijackers

In this work, we use the term hijacking to describe when viral proteins disrupt host processes (Davey *et al*, 2011). We expect that SARS-CoV-2 will hijack many human proteins (Gordon *et al*, 2020); however, our study found direct structural evidence of hijacking involved only ~7% of the viral proteome (Fig 3), comprising NSP1, NSP3, spike glycoprotein, envelope protein, and ORF9b protein (Dataset EV5).

NSP1 has been reported to hijack the small ribosomal subunit (40S) by blocking entry and translation of host mRNA (Schubert

et al, 2020), greatly altering the host cell transcriptome (Yuan et al, 2020b), and effectively blocking translation of antiviral defense factors, such as DDX58 (a.k.a. retinoic acid-inducible gene I) or IFNB1 (Thoms et al, 2020). However, the translation of viral mRNA is facilitated via interactions between the 5′ untranslated region (5′ UTR) with the N-terminal region of NSP1 (preprint: Shi et al, 2020). The 14 matching structures showing these hijacking mechanisms may aid structure-based drug design of anti-COVID-19 therapeutics targeting NSP1 or the 5′ UTR of viral mRNA.

NSP3 is also implicated in suppressing host RNA translation and in enhancing viral RNA translation. This is reported to involve hijacking of 40S by a ternary complex of NSP3 with PAIP1 and PABP; however, to date, the only available structural evidence (Lei et al, 2021) shows PAIP1 in complex with the SUD-N region of SARS-CoV NSP3 (Fig 3B). PAIP1 is known to regulate translation initiation of mRNAs containing a poly(A) tail (Grosset et al, 2000), which is believed to be characteristic of coronaviral transcripts (Lai & Stohlman, 1981).

NSP3 is also believed to cleave ubiquitin-like domains from host proteins, thereby suppressing innate immune responses and disrupting proteasome-mediated degradation (Lei et al, 2018). We found matching structures suggesting that NSP3 may bind to three of the four human ubiquitin proteins (UBB, UBC, and UBA52). These structures further suggest that each NSP3 molecule can bind up to two ubiquitin-like domains (Fig 3C), so may also reverse polyubiquitination. We also found structures showing binding to the ubiquitin-like domains of ISG15 (a.k.a. interferon-stimulated gene 15), which attach to newly synthesized proteins; ISGylation does not induce degradation but is thought to disturb virion assembly, so reversing this modification may be necessary for viral replication (Lei et al, 2018).

Spike glycoprotein is known to bind receptors on the host cell membrane, thereby initiating membrane fusion and viral entry (Hoffmann et al, 2020). Multiple matching structures showed details on the hijacking of ACE2, a carboxypeptidase that normally cleaves vasoactive peptides. Two of these structures also show binding to SLC6A19 (a.k.a. B⁰AT1; Fig 3D), an amino acid transporter that interacts with ACE2 (Camargo et al, 2009); these structures reveal additional molecular mechanisms underlying spike glycoprotein entry (Yan et al, 2021b). Entry may also be facilitated by hijacking of another cell surface protein called NRP1 (a.k.a. neuropilin-1; Daly et al, 2020).

Two additional matching structures showed hijacking of the membrane receptor DPP4 (a.k.a. "Dipeptidyl peptidase IV" or CD26) by MERS-CoV spike glycoprotein (Dataset EV5). It has been speculated that DPP4 may also be used by SARS-CoV-2 to enter host immune cells (Radzikowska et al, 2020), although this speculation is not supported by in vitro studies (Tai et al, 2020) or by analysis of the matching structures (Appendix Fig S1B). We concluded that current evidence does not support binding between DPP4 and spike glycoprotein from SARS-CoV-2, so we did not include this interaction in Figs 1 and 3.

The viral envelope protein is believed to hijack MPP5 (a.k.a. PALS1), preventing it from performing its normal role in intracellular tight junctions and thereby driving lung epithelium disruption in coronavirus infection (Teoh et al, 2010). Currently, this hijacking is captured in only one structure (Table 1) that lacks a supporting

scientific publication and matches only a nine-residue region of E protein (Fig 3F).

ORF9b protein hijacking of the outer mitochondrial membrane protein TOMM70 (Fig 3G) is reported to be one of the key viral-mitochondrial interactions occurring during infection; however, much about this interaction remains unclear, and currently, it is captured in only one structure (Gordon et al, 2020).

## Teams

We found structural evidence of interaction between viral proteins for ~29% of the viral proteome (Fig 4), comprising eight SARS-CoV-2 proteins; these proteins divided into two disjoint teams, described below.

Team 1 comprised NSP7, NSP8, NSP9, NSP12, and NSP13, all members of the viral replication and translation complex (Fig 4A). NSP12 alone can replicate RNA, as can NSP7 + NSP8 acting together (te Velthuis et al, 2012). However, replication is greatly stimulated by cooperative interactions between the RTC proteins (Kirchdoerfer & Ward, 2019). Using an assembly matrix layout revealed that, of the many possible heteromeric combinations, only a small number were observed among the >200 structures matching these proteins (Fig 4A). The matrix also suggests the order in which RTC components may assemble, and, by omission, makes clear that several proteins implicated in genome replication (Subissi et al, 2014) are missing (NSP3, NSP10, NSP14, and NSP16). These outcomes demonstrate the value of systematically modeling all available structural states, rather than only one or few states per protein.

Team 2 comprised NSP10, NSP14, and NSP16 (Fig 4B). All matching structures found for NSP14 and many found for NSP16 showed binding with NSP10, consistent with the belief that NSP10 is required for NSP16 RNA-cap methyltransferase activity (Decroly et al, 2011), and also for NSP14 methyltransferase and exoribonuclease activities (Ma et al, 2015). NSP10 was found to form a homododecamer, and all three observed oligomeric states involving NSP10 share a common binding region (Fig 4B), consistent with previous reports (Bouvet et al, 2014). This, together with the four heteromeric states observed among the >100 matching structures involving these proteins, suggested that NSP10, NSP14, and NSP16 interact competitively—in contrast to the mostly cooperative interactions seen in team 1. We speculate that NSP10 may be produced at higher abundance than NSP14 or NSP16, as it could otherwise be rate limiting for viral replication.

Finally, it is noteworthy that no interactions were found between the 12 virion or accessory proteins (Fig 1, bottom third), many of which are known to assemble to form the mature virus particle. This, again, highlights limitations in currently available structural data.

## Suspects

This leaves 14 of the 27 viral proteins in a final category we call suspects (Dataset EV7): These are proteins thought to play key roles in infection, but having no structural evidence of interaction with other proteins (viral or human). We divided the suspects into two groups, based on matching structures.

Group 1 suspects were those with at least one matching structure: NSP4, NSP5, NSP15, ORF3a, ORF7a, and ORF8, and nucleocapsid protein. Some of these have been well studied (e.g., NSP5 had 450 matching structures). Yet none of these proteins had significant similarity to any experimentally determined 3D structure involving human proteins or to any structure showing interactions between viral proteins—based on the methods used in this work.

Group 2 suspects were those with no matching structures: NSP2, NSP6, matrix glycoprotein, ORF6, ORF7b, and ORF9c, and ORF10. These are structurally dark proteins (Perdigão *et al*, 2015), meaning not only is their structure unknown, but also that they have no significant sequence similarity to any experimentally determined 3D structure—based on the methods used in this work. Thus, these are the worst characterized viral proteins from a structural perspective (Fig 1, Dataset EV7). These proteins are ripe candidates for advanced modeling strategies, e.g., using predicted residue contacts combined with deep learning (Senior *et al*, 2020).

### Structural coverage

In combination, these dark proteins and all dark regions found in our analysis accounted for 31% of the viral proteome (Dataset EV8); this was somewhat lower than the 54% average darkness found across all viral proteomes in SwissProt (Perdigão *et al*, 2015), indicating that coronaviruses are comparatively well studied. Only 3.9% of the dark proteome was predicted to be disordered, compared with 2.0% for the non-dark proteome (Dataset EV8). Thus, disorder did not account for the majority (96%) of the dark proteome, which remains largely unexplained, consistent with previous observations (Perdigão *et al*, 2015).

Within the non-dark or modellable fraction of the proteome (69%), a total of 26 CATH superfamilies assignments were found. We note that exactly the same 26 CATH assignments were also found in the proteome of SARS-CoV (Dataset EV9). Furthermore, based on PredictProtein, the SARS-CoV-2 proteome was also found to have a very similar distribution for predicted secondary structure content, compared to both SARS-CoV and MERS-CoV (Dataset EV12).

In both SARS-CoV and SARS-CoV-2 proteomes, the most common topology was the Rossmann fold, which had five recurrences (in NSP3, NSP15, NSP16, and twice in NSP13), followed by

the alpha-beta plait and macro-like topologies—each with three recurrences. An additional five topologies each had two recurrences (ruvA helicase-like, thrombin-like, ubiquitin-like, jelly roll, and heptad repeat). These recurring topologies could be grouped into three broad functional categories: RNA interaction (especially unwinding), protein modification (removal of ADPr or of ubiquitin-like attachments), and protein oligomerization. Of the five remaining, non-recurring topologies, three also shared these functions; this left a final two topologies, glutaredoxin-like and immunoglobulin-like, that belonged to two additional functional categories—redox metabolism and immune dysregulation, respectively (Dataset EV9). Identifying these five broad functional categories and eight recurring topologies may help focus future research efforts on understanding the molecular mechanisms of the viral proteome, and on developing antiviral drugs.

Finally, we note that the 26 CATH superfamilies covered only 33% of the total proteome, thus leaving 36% of the proteome with structural information that could not be assigned to existing topologies based on the current CATH library (version 4.3). These unassigned structural regions are ripe candidates for further structural characterization.

### Conclusions

In conclusion, we have assembled a wealth of information, not available from other resources, about the structure of the viral proteome. Our analysis of these data has provided insight into how viral proteins self-assemble, how NSP3 and NSP13 may mimic human proteins, and how viral hijacking reverses post-translational modifications, blocks host translation, and disables host defenses. In addition, our study helps direct future research by quantifying and drawing attention to aspects of the viral proteome that remain unknown; this includes regions with unknown structure, as well as regions with known structure but unknown function. These outcomes are visually summarized in the structural coverage map (Fig 1), a novel layout concept that not only provides an insightful overview of available structural evidence, but can also be used as a navigation aid, helping researchers find and explore 3D models of interest. The resulting Aquaria-COVID resource (https://aquaria.ws/covid) aims to fulfill a vital role during the ongoing COVID-19 pandemic, helping scientists use emerging structural data to understand the molecular mechanisms underlying coronavirus infection.

# Materials and Methods

### Reagents and Tools table

| Resource | Reference or source | Identifier or version number |
|---|---|---|
| **SARS-CoV-2 protein sequences** | | |
| Polyprotein 1a | UniProt | P0DTC1 |
| Polyprotein 1ab | UniProt | P0DTD1 |
| Spike glycoprotein | UniProt | P0DTC2 |
| ORF3a protein | UniProt | P0DTC3 |

**Reagents and Tools table** (continued)

| Resource | Reference or source | Identifier or version number |
|---|---|---|
| Envelope protein | UniProt | P0DTC4 |
| Matrix glycoprotein | UniProt | P0DTC5 |
| ORF6 protein | UniProt | P0DTC6 |
| ORF7a protein | UniProt | P0DTC7 |
| ORF7b protein | UniProt | P0DTD8 |
| ORF8 protein | UniProt | P0DTC8 |
| Nucleocapsid protein | UniProt | P0DTC9 |
| ORF9b protein | UniProt | P0DTD2 |
| ORF9c protein | UniProt | P0DTD3 |
| ORF10 protein | UniProt | A0A663DJA2 |
| **Software** | | |
| HH-suite | https://github.com/soedinglab/hh-suite/releases/tag/v3.3.0 | 3.3.0 (ac765987bd) |
| HMMER | http://hmmer.org/ | 3.3 |
| cath-resolve-hits | https://cath-tools.readthedocs.io/en/latest/ | v0.16.2-0-ga9f860c |
| CATH API | https://github.com/UCLOrengoGroup/cath-api-docs | 4.3 |
| PredictProtein API | https://api.predictprotein.org/v1/results/molart/P0DTC1 | |
| SNAP2 API | https://rostlab.org/services/aquaria/snap4aquaria/json.php?uniprotAcc=P0DTC1 | |
| PSSH2 tools | https://github.com/aschafu/PSSH2.git | |
| Jolecule | https://jolecule.com/ | |
| Aquaria | https://aquaria.ws/ | |
| **Other** | | |
| UniRef30 | http://wwwuser.gwdg.de/~compbiol/uniclust/2020_03/UniRef30_2020_03_hhsuite.tar.gz | 2020_03 |
| PDB | https://ftp.rcsb.org/pub/pdb/data/structures/divided/mmCIF | 27 March, 2021 |
| CATH-Gene3D FunFamsHMM library | ftp://orengoftp.biochem.ucl.ac.uk/cath/releases/all-releases/v4_3_0 | 4.3 |
| CATH nr40 | ftp://orengoftp.biochem.ucl.ac.uk/cath/releases/all-releases/v4_3_0 | 11 September, 2019 |
| PSSH2 | https://doi.org/10.5281/zenodo.4279163 | 27 June, 2020 |

## Methods and Protocols

### SARS-CoV-2 sequences

This study was based on the 14 protein sequences provided in UniProtKB/SwissProt (downloaded March 11, 2021; https://www.uniprot.org/statistics/) as comprising the SARS-CoV-2 proteome (Reagents and Tools table). SwissProt provides polyproteins 1a and 1ab (a.k.a. PP1a and PP1ab) as two separate entries, both identical for the first 4,401 residues; PP1a then has four additional residues ("GFAV") not in PP1ab, which has 2,695 additional residues not in PP1a. SwissProt also indicates residue positions at which the polyproteins become cleaved into protein fragments, named NSP1 though NSP16. The NSP11 fragment comprises the last 13 residues of PP1a (4,393–4,405). The first 9 residues of NSP12 are identical to the first nine of NSP11, but the rest of that 919 residue long protein continues with a different sequence due to a functionally important frameshift between ORF1a and ORF1b (Nakagawa *et al*, 2016). Thus, following cleavage, the proteome comprises a final total of 27 separate proteins.

### Sequence-to-structure alignments

The 14 SARS-CoV-2 sequences were then mapped onto all related 3D structures using the PSSH2 tools, which run the Aquaria sequence-to-structure processing pipeline (O'Donoghue *et al*, 2015). As a first step in this process, each sequence was systematically compared with sequences derived from all 176,388 available PDB entries (downloaded March 27, 2021). These comparisons used HHblits v3.3.0 and UniRef30 (components of HH-suite; Steinegger *et al*, 2019) in the processing pipeline defined previously (O'Donoghue *et al*, 2015), accepting all sequence-to-structure alignments with a significance threshold $E \leq 10^{-10}$ or with a pairwise identity $\geq 90\%$. The resulting set of sequence-to-structure alignments for SARS-CoV-2 (Data ref: Schafferhans *et al*, 2021) was added to PSSH2, a database with over 100 million sequence-to-structure alignments, covering all SwissProt sequences (Data ref: Schafferhans & O'Donoghue, 2020).

Each time a user visits an Aquaria web page corresponding to a viral protein, Aquaria performs the following steps:

1   The UniProt primary accession is used in a database query to find all exactly matching chains in the latest version of the PDB

(updated weekly), using the sequence cross-references given in each PDB entry. Sequence-to-structure alignments are then created based on the information provided in each PDB entry.

2    In addition, the UniProt primary accession is converted to an MD5 hash generated from the corresponding protein sequence; the MD5 hash is then used to return a list of all related PDB chains and sequence-to-structure alignments stored in PSSH2.

3    The results retrieved from (1) and (2) are processed to merge any duplicates, by checking whether the alignments from the PDB record and from PSSH2 overlap, using the criteria previously described (O'Donoghue *et al*, 2015).

4    In the case of polyprotein 1a and 1ab, we additionally merged matches that occurred in the overlapping regions of sequence (NSP1-NSP10); this ensures that the same number of matching structures is shown in these regions for both proteins.

5    The final, merged set of sequence-to-structure alignments are then clustered as described previously (O'Donoghue *et al*, 2015).

The final counts for matching structures shown on the Aquaria interface, and used in this work, are based on distinct PDB chain entries. This means that whenever one viral protein sequence matched to two duplicate chains occurring the same PDB entry, this was counted as only one distinct matching structure; however, whenever two viral protein sequences matched to two distinct chains in one PDB entry, this was counted as two distinct matching structures.

For each sequence-to-structure alignment derived from HHblits, the Aquaria interface shows the pairwise sequence identity score, thus providing an intuitive indication of how closely related the given region of SARS-CoV-2 is to the sequence of the matched structure. However, to more accurately assess the quality of the match, Aquaria also gives an E-value, calculated by comparing two hidden Markov models (HMMs), one generated for each of these two sequences. Note that these E-values depend on current knowledge and, in some cases, can change dramatically as new sequences or structures become available.

All SARS-CoV-2 matching structures derived in this work can be directly accessed from links provided in Datasets EV1–EV3; additionally, the underlying SARS-CoV-2 sequence-to-structure alignments are available online for download (Data ref: Schafferhans *et al*, 2021), as is the full PSSH2 database (Data ref: Schafferhans & O'Donoghue, 2020). In Datasets EV1–EV3, alignments derived entirely from PDB annotations are indicated via a null E-value ("-"), to distinguish them from PSSH2-derived alignments. Note that PSSH2 alignments were generated based on the full-length protein sequence used in the experiment to derive each PDB entry ("SEQRES"); this sequence often includes regions that are not visible in the final structure due to lack of data (typically occurring in regions with intrinsic disorder).

As indicated above, all new PDB structures are automatically imported into Aquaria each week. Thus, structures determined for SARS-CoV-2 proteins are available via the Aquaria web interface within a week after they are released. Updates to the PSSH2 entries that map from SARS-CoV-2 sequences to structures from related organisms are planned quarterly, while updates for the complete PSSH2 database are planned annually.

### Alignment E-value threshold

In this work, we have used an HHblits *E*-value of $10^{-10}$ as the primary threshold for predicting structure based on sequences. This threshold was derived from a detailed assessment of specificity and sensitivity of structure predictions (O'Donoghue *et al*, 2015), in which two structures were assessed to be similar if they had $\geq 30\%$ structural overlap, as measured by COPS (Frank *et al*, 2010). According to this assessment, using $E \leq 10^{-10}$ should result in a 1% false positive rate and 83% recovery rate. However, in the 5 years since this previous assessment was done, HHblits has been substantially updated; thus, for this work we decided to do a preliminary re-assessment of these benchmarks. Unfortunately, COPS has since been discontinued; thus, for benchmarking accuracy and precision, we used CATH instead (Sillitoe *et al*, 2021). Our test data set comprised 23,028 sequences from the CATH nr40 data set. We built individual sequence profiles against UniClust30 and used these profiles to search against "PDB_full", a database of HMMs for all PDB sequences. We then evaluated how many false positives were retrieved at an E-value lower than $10^{-10}$, where a false positive was defined to be a structure with a different CATH code at the level of homologous superfamily (H) or topology (T). We compared the ratio of false positives received with HH-suite3 and UniClust30 (Steinegger *et al*, 2019) with a similar analysis for data produced in 2017 with HH-suite2 and UniProt20, and found that in both cases the false positive rate was 2.5% at the homology level (H), and 1.9% at the topology level (T). The recovery rate, i.e., the ratio of proteins from the CATH nr40 data (with < 40% sequence identity) found by our method that have the same CATH code, was slightly higher with HH-suite3 (20.8% vs. 19.4%). Differences in benchmark values, compared with our 2015 assessment, are expected, due to many differences between CATH and COPS. Given the rather close similarity in CATH-based values based on HH-suite2 and HH-suite3, we concluded that the chosen E-value cutoff is still valid.

### PredictProtein features

To facilitate analysis of SARS-CoV-2 sequences, we enhanced the Aquaria resource to include PredictProtein features (Yachdav *et al*, 2014), thus providing a very rich set of predicted features for all Swiss-Prot sequences. The five PredictProtein feature sets used in this work were fetched via:

https://api.predictprotein.org/v1/results/molart/P0DTC1

#### Conservation

The first PredictProtein feature set is generated by ConSurf (Celniker *et al*, 2013) and gives, for each residue, a score between 1 and 9, corresponding to very low and very high conservation, respectively. These scores estimate the evolutionary rate in protein families, based on evolutionary relatedness between the query protein and its homologues from UniProt using empirical Bayesian methods (Mayrose *et al*, 2004).

#### Disorder

This feature set gives consensus predictions generated by Meta-Disorder (Schlessinger *et al*, 2006), which combines outputs of several structure-based disorder predictors to classify each residue as either disordered or not disordered.

#### Flexibility

This feature set predicts, for each residue, normalized B-factor values expected to be observed in an X-ray-derived structure, generated by PROFbval (Schlessinger *et al*, 2006). For each residue,

PROFbval provides a score between 0 and 100; a score of 50 indicates average flexibility, while ≥ 71 indicates highly flexible residues.

### Solvent accessibility

This feature set gives a two-state prediction for each residue—either buried or exposed to the solvent—generated by RePROF (Yachdav et al, 2014).

### Topology

This feature set is generated by TMSEG (Bernhofer et al, 2016), a machine learning model that uses evolutionary-derived information to predict regions of a protein that traverse membranes, as well as the subcellular locations of complementary (non-transmembrane) regions.

For the first four of these feature sets, we also used the Predict-Protein API to calculate average values for each of the final 27 viral proteins (Dataset EV10). For NSP1-NSP10, the average values were generated using the polyprotein 1a sequence; for NSP12-NSP16, the polyprotein 1ab sequence was used. In addition, we calculated average values for each of the NSP3 regions (Dataset EV11).

Finally, to facilitate a balanced comparison of the SARS-CoV-2 proteome with that of SARS-CoV and MERS-CoV (Dataset EV12), we also used the PredictProtein API to fetch secondary structure predictions from RePROF (Yachdav et al, 2014).

### SNAP2 features

We further enhanced Aquaria to include SNAP2 (Hecht et al, 2015) features, which provide information on the mutational propensities for each residue position. In Aquaria, two SNAP2 features sets are fetched via:

https://rostlab.org/services/aquaria/snap4aquaria/json.php?uniprotAcc = P0DTC1

### Mutational sensitivity

The first SNAP2 feature set provides, for each residue position, a list of 20 scores that indicate the predicted functional consequences of the position being occupied by each of the 20 standard amino acids. Large, positive scores (up to 100) indicate substitutions likely to have deleterious changes, while negative scores (down to −100) indicate no likely functional change. From these 20 values, a single summary score is calculated based on the total fraction of substitutions predicted to have deleterious effect, taken to be those with a score > 40. The summary scores are used to generate a red to blue color map, indicating residues with highest to least mutational sensitivity, respectively.

### Mutational score

The second SNAP2 feature set is based on the same 20 scores above, but calculates the single summary score for each residue as the average of the individual scores for each of the 20 standard amino acids.

### UniProt features

UniProt features are curated annotations, and therefore largely complement the automatically generated PredictProtein features. In Aquaria, for each protein sequence, the UniProt feature collection is fetched via:

https://www.uniprot.org/uniprot/P0DTC1.xml

### CATH features

Unfortunately, the SARS-CoV-2 protein sequences (Reagents and Tools Table) are not included in the current production release of CATH (version 4.3; Sillitoe et al, 2021). Thus, for this work, we used the resources used in the CATH database generation workflow to create a pre-release version of CATH assignments for these sequences.

For each sequence, CATH superfamily and functional family (FunFams) assignments were obtained by running HMMER (Mistry et al, 2013) and cath-resolve-hits (Lewis et al, 2019) against the CATH-Gene3D v4.3 FunFamsHMM library. Superfamilies are regions of evolutionarily related protein sequences that are predicted to have similar 3D structures and to share general biological functions. FunFams further partition each superfamily into subsets expected to have specific biological functions in common (Sillitoe et al, 2012).

Each CATH superfamily has a unique four-integer identifier, of which the first three integers identify a collection of superfamilies that share a common structural topology or fold, but do not have an evolutionary relation. In this work, we used topologies assigned via the first three CATH identifiers to assess the recurrence of folds in the viral proteome.

Dataset EV9 gives a summary of the resulting CATH superfamily and FunFam assignments for SARS-CoV-2 and related viruses; further details are available online (Data ref: Bordin, 2021). These assignments are planned to be integrated in the next release of CATH (version 4.4), currently scheduled for early 2022.

In addition, we enhanced the Aquaria resource so that, whenever a user loads a SARS-CoV-2 sequence, the above superfamily and FunFam assignments are fetched. For all other sequences in SwissProt, these assignments are fetched from the CATH API: https://github.com/UCLOrengoGroup/cath-api-docs

We further enhanced the Aquaria interface so that, whenever a user hovers over a representation of a superfamily or FunFam assigned to a protein sequence, the CATH API is also used to gather related data. These data are shown in a popup using compact, interactive visualizations that give access to detailed information on the biological function, and phylogenetic distribution of proteins with the specified superfamily or FunFam.

### Aquaria core

For this work, the Aquaria core codebase has been substantially refactored. We changed all client-server data exchanges to use GET requests, instead of web sockets as used previously (O'Donoghue et al, 2015). This allowed us to further implement both server-side caching and client-side caching, resulting in greatly improved performance. Another major change was to the user interface, where we removed the previously used Java applet (O'Donoghue et al, 2015), and replaced it with Jolecule, a JavaScript-based molecular graphic component (https://jolecule.com/), that we further augmented to enable feature mapping. In this work, Jolecule was used to determine the set of residues comprising intermolecular contacts by selecting all residues for one protein, then applying Jolecule's "Neighbours" function. This highlights all residues in which any atom is within 5 Å.

### Structure coverage matrix

We created a web page featuring a matrix view of the 14 UniProt sequence and showing the total number of matching structures

found. This page allows navigation to the corresponding Aquaria page for each protein sequence (https://aquaria.ws/covid#Matrix).

### Structural coverage map

For each contiguous region of the viral proteome with matching structures, we selected a single representative structure (Fig 1); in most cases, this was based on identity to the SARS-CoV-2 sequence; however, in some cases, we chose structures showing the simplest or most common biological assembly. Under the name of each viral protein, the total number of matching structures found is indicated (see Sequence-to-Structure Alignments, above). Below each structure, a tree graph indicates structural evidence of mimicry (i.e., where the viral sequence aligns onto human proteins; Dataset EV4), hijacking (i.e., where viral proteins directly bind to human proteins; Dataset EV5), or other types of binding (i.e., where viral proteins bind antibodies, HLA, RNA, inhibitory peptides, or other viral proteins; Dataset EV6).

Each state shown in these tree graphs was derived from an automated analysis that listed all molecules present in each matching structure. This defined lists of putative states (Datasets EV4–EV6), which were then manually assessed by visually examining relevant structures, and by reading source literature. For brevity, we excluded matching structures showing evidence of mimicry or hijacking of proteins from other (non-human) host organisms; these matching structures are, however, included in Datasets EV1–EV3. Many of the listed putative states were supported by direct evidence (i.e., at least one structure determined for SARS-CoV-2). For all putative states with only indirect evidence, each manual assessment is summarized in the Results, and the final outcomes are indicated in Datasets EV4–EV6. For a small number of putative states, we assessed that there was insufficient evidence to conclude that they occur in SARS-CoV-2; two such cases are highlighted in Appendix Fig S1.

Figure 1 also indicates dark and non-dark regions of the proteome, derived by merging all sequence-to-structure alignments, and also accounting for structural gaps arising from lack of data. These regions are specified in Dataset EV8, which also provides a comparison of disorder in dark versus non-dark regions, using PredictProtein's Meta-Disorder service (Schlessinger *et al*, 2006).

In addition to summarizing structural evidence, we designed Fig 1 to help researchers find specific structural states they may be interested in. Each of the states represented in the coverage map is hyperlinked to representative matching structures, listed in Table 1. These links can be accessed via a stand-alone PDF version of Fig 1 hosted online at our associated Aquaria-COVID resource (https://aquaria.ws/covid).

### Assembly matrix

To visually summarize the many structural matches that contain heteromers, we devised a novel matrix-based layout (Fig 4). Each structure was first automatically analyzed to determine the identity and number of macromolecules present. This analysis resulted in a list of all distinct heteromeric states observed among the matching structures. For each viral interaction team, we found key molecules that occurred in most states (NSP12 and RNA for team 1 and NSP10 for team 2). We used these key molecules to define columns in a matrix that summarizes all observed

states, with each edge between adjacent cells indicating a potential assembly step. For brevity, we omitted most monomers and homomers. Each final assembly matrix (Fig 4) shows that only a small subset of all combinatorial possible states was observed and provides insight into the order in which heteromers may assemble.

## Data availability

The datasets and computer code produced in this study are available in the following databases:

- CATH assignments for SARS-CoV-2, SARS-CoV, and MERS-CoV: Zenodo 4915950 (https://doi.org/10.5281/zenodo.4915950)
- Aquaria code: GitHub (https://github.com/ODonoghueLab/Aquaria/releases/tag/v1.0)
- Sequence-to-structure alignments and non-dark regions for SARS-CoV-2: Zenodo 4934861(https://doi.org/10.5281/zenodo.4934860)

**Expanded View** for this article is available online.

### Acknowledgements

Thanks to Tim Mercer and Giulia Wang (Garvan Institute, Australia), Phil Austin (University of Sydney, Australia) and Lucy van Dorp (UCL Genetics Institute, UK) for helpful feedback and discussions, to Ian Sillitoe (UCL, UK) for helpful advice regarding the CATH API, to Tim Karl, Michael Bernhofer, and Maria Littmann (TUM, Germany) for advice regarding the PredictProtein server. We are grateful to Max Ott (CSIRO, Australia) for advice on improving the performance and reliability of the Aquaria web application. SIOD and NS were supported by the Garvan Research Foundation, Tour de Cure Australia, and Sony Foundation Australia. JP was supported by the Wellcome Trust (218259/Z/19/Z). NB acknowledges financial support from the Biotechnology and Biological Sciences Research Council (BBSRC; https://doi.org/10.13039/501100000268) project number BB/R009597/1. CD was supported by the Bundesministerium für Bildung und Forschung (BMBF; https://doi.org/10.13039/501100002347), project numbers 031L0168 and 01IS17049.

### Author contributions

The authors have been listed in order of contribution. SIOD designed this study and led in coordinating co-author contributions, in data analysis, in manuscript writing, and in figure preparation. He also participated in the database generation. AS led in 3D model generation, in PSSH2 validation, and was involved in data analysis and in writing the manuscript. NS coordinated integration of Aquaria code improvements and was involved in data analysis, in writing the manuscript, and in creating the tables. CS led in the design of graphical user interface elements and contributed to figure generation. SK led in implementing CATH, SNAP2, and PredictProtein features into Aquaria, participated in data analysis, and assisted in creating the tables. BKH led the integration for Jolecule into the Aquaria user interface. SA and MA implemented key new features to the molecular graphics components. JBP contributed to data analysis and provided strategic input into the manuscript. CD helped with integration of SNAP2 and PredictProtein features into Aquaria and was involved in data analysis, in writing the manuscript, and in creating the tables. NB participated with the implementation of CATH features into Aquaria and assisted in creating the tables. BR contributed to model generation, oversaw SNAP2 and PredictProtein developments, and gave strategic input into the manuscript.

## Conflict of interest

The authors declare that they have no conflict of interest.

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
