## [Review Process File · Molecular Systems Biology]

SARS-CoV-2 structural coverage map reveals viral protein assembly, mimicry, and hijacking mechanisms

Seán O'Donoghue, Andrea Schafferhans, Neblina Sikta, Christian Stolte, Sandeep Kaur, Bosco Ho, Stuart Anderson, James Procter, Christian Dallago, Nicola Bordin, Matt Adcock, and Burkhard Rost
DOI: 10.15252/msb.202010079

Corresponding author(s): Seán O'Donoghue (sean@odonoghuelab.org)

Review Timeline:

Submission Date:	24th Oct 20
Editorial Decision:	7th Jan 21
Revision Received:	4th Jul 21
Editorial Decision:	29th Jul 21
Revision Received:	5th Aug 21
Accepted:	6th Aug 21

Editor: Maria Polychronidou

Transaction Report:

Thank you again for submitting your work to Molecular Systems Biology. I would like to apologise for the exceptional delay in getting back to you with a decision, which was due to the fact that after repeated reminders we still have not received the comments of reviewer #3. The reason why I have waited for them until now is that they repeatedly promised to deliver their comments, and I felt that it would have been constructive to have a third opinion from an expert in the field. Nevertheless, to not delay the process any further, I have now decided to proceed with making a decision based on the two available reports. Overall, the reviewers acknowledge that the presented database seems useful. However, they raise a series of concerns, which we would ask you to address in a major revision.

Without repeating all the points listed below, some of the more fundamental points are the following:

- Reviewer #1 mentions that a more detailed report of structural coverage needs to be provided.
- Reviewer #1 points out that the database would need to be regularly updated in an automated manner in order to not become obsolete. This is particularly important as this is a field of active research and the database needs to remain relevant longterm.
- As reviewer #1 recommends, the claims regarding PPIs and protein complexes need to be better supported or toned down.

All issues raised by the referees would need to be satisfactorily addressed. Please let me know in case you would like to discuss in further detail any of the issues raised. As you might already know, our editorial policy allows in principle a single round of major revision so it is essential to provide responses to the reviewers' comments that are as complete as possible.

On a more editorial level, we would ask you to address the following point.

Reviewer #1:

In this manuscript the authors have provided hundreds of structural models for SARS-CoV-2 proteins and integrated them for visualization in an online resource database, which is aesthetically pleasant and intuitive to use and analyze.

The major novelty of this work is the systematic structural coverage of the viral proteins and their online map, which can be used by researchers for further investigations on the nature of the proteins. Minor novelties include, but are not limited to, efforts to classify functional aspects of proteins based on their structural similarities with those deposited in the protein data bank and extensive reprogramming of available tools and their integration with major structure and sequence analysis tools.

The paper is well-written. Personally, I find the paper interesting and of interest to the scientific community in general. However, I have four major comments for the authors, which I list below, and must be addressed in a revised form before publication:

1. The structural coverage claims are misleading: Of course, 872 models are quite a lot, and I congratulate the authors for finding that many. But there is no analysis or statistics on how much sequence do these models finally cover over the whole viral proteome. There are some insights here and there about absence of structural homology, but no systematized, global evaluation of "modellable" and "unmodellable" regions. Therefore, I invite the authors to properly report on the complete structural coverage by answering, for example, the following questions:
 - a. How much of the total proteome is covered by the 872 models, and which proteins are the most/less studied?
 - b. What is the distribution of homology across the total proteome?
 - c. The regions that do not have any homology yet: Is it because of high disordered content or another factor?
 - d. What is the overall content in secondary structure in this proteome as compared to other (viral) proteomes?
 - e. Are the folds of SARS-CoV-2 proteins redundant (e.g. found more than 1 time among SARS-CoV-2 proteins) or rare as compared to other (viral) proteomes?
2. The database has to be automatically updateable. Due to the COVID-19 pandemic, more structures are expected to be deposited in the PDB (or even modelled by other groups). Reported results, in a few months, will render the database obsolete, and of no use to researchers. Therefore, automatically updating the structural coverage of all proteins in a weekly manner and including links to other related resources will make the database immensely useful for the community and a solid website for researchers to find out the latest structural coverage updates for SARS-CoV-2 proteins.
3. The database has to be publicly shared and researchers have to be able to completely download its components. The most straightforward way that this can be achieved is by providing the database for download and installation on researchers' own host machines.
4. Novelty aspect of the manuscript has to be re-evaluated: Here, the authors have included two rather daring aspects which I highly suggest toning down.
 - a. The title: There is no evidence in a computational homology-based work for disruption of host immunity. Computational work is critical, but it is based on data collection. This has been very nicely done in this manuscript. In addition, it is based on analysis, which has also been performed accurately in this work. However, claims of relationships between biological phenotypes and data-mined models have to be supported by experimental data. Therefore, I invite the authors to change their title pointing to the strengths of their study, those being the (a) structural coverage map, (b) the visualization and open-source nature of their project, (c) the updateability of their data, as I

suggested in (1), and (d) the direct homology-based comparisons that provide hypotheses on protein function but not on cellular effects, which are much more complex.

b. There is no modeling of protein-protein interactions for the viral proteins anywhere in this paper. This is not necessarily bad; the authors focus on single proteins and it is fine. However, if the authors do not perform this, then all claims in the manuscript regarding complexes have to be down toned or removed, because these come from authors' individual observations and not from a systematized study of protein-protein interactions of viral proteins. This includes, but is not limited to, Fig. 4B, which I suggest to completely remove: Structure is known, modeling has been done by several groups already (also MD), glycans are missing, conformation changes are not discussed, insights into binding intermediates are not presented, loop of RBD domain is not systematically described and how residues affect the interaction etc. Just showing a Figure panel of the complex is out of the scope of this paper, to my opinion.

I am convinced that after addressing the above-mentioned revisions, which I consider not to be time-consuming, that the manuscript and the web will become a central hub for researchers and additional arsenal to fight against the COVID-19 pandemic.

Reviewer #2:

This is an interesting and nicely written paper describing a new database of structural properties for the 14 proteins found in the proteome of the COVID-19 virus. The database is an extension of the Aquaria database for exploring structural properties of proteins. The authors provide a large collection of 3D protein structures (minimal models) from PDB that have significant sequence similarity with COVID-19 proteins. The database provides a very good and user-friendly interface for visually exploring these structures and the regions of similarity between each model and the COVID-19 protein. This interface also provides all protein structural annotation features that are readily available through the Aquaria server. The authors described with many examples in the paper how this database can be used to obtain insights at multiple levels into the behaviour of COVID-19 proteins inside the host cell. I find this database to be very useful for the study of COVID-19. I have only a few minor comments.

1 - How do the authors discriminate mimicry from hijacking? What I know is that protein mimicry is a molecular feature that drives hijacking. It seems to me that they consider a viral protein to mimic a host protein, but not hijack a process, if none of the structures mapping to the viral protein include an interaction partner from the host? If that is true, I think it would be better to make this definition clear at the beginning of the Discussion before they start talking about the mimicry and hijacking groups, where they leave it for the readers to conclude on their own.

2 - The authors state on Page 9 regarding the rare cases they found for viral mimicry or hijacking that "This may indicate that host interactions are rarely used in COVID-19 infection, consistent with the notion that viral activity is largely shielded from the host. However, other experimental techniques have found many more interactions between viral proteins (Pan et al., 2008), and with host proteins (Gordon et al., 2020). Thus, the small number of interactions found in this work likely indicates limitations in current structural data."

This brings to my attention a couple of computational studies that have shown evidence of viral mimicry and hijacking at the larger interactome scale, using sequence/structural similarity approaches at the domain level as well as the residue level between viral proteins and human

proteins. For example, this prior study: <https://www.pnas.org/content/108/26/10538>.

While it is plausible that the rare cases of mimicry or hijacking found here may be either unique to this virus, or due to limitations in current structural data, an alternative hypothesis that needs to be ruled out is the specific criteria used here by the authors to identify homologous structures. How robust are the conclusions with regard to variations in these criteria? I recommend that the authors at least address this point in their discussion.

3 - The notion of "feature sets" used throughout the paper may be confusing to some readers. As it first sounds like there are >32,000 unique features that are imported by the Aquaria server, but later in the methods it appears that these 32,000 features are more like values of a few (but significant) number of features.

REVIEWER #1

1. *'The structural coverage claims are misleading: Of course, 872 models are quite a lot, and I congratulate the authors for finding that many. But there is no analysis or statistics on how much sequence do these models finally cover over the whole viral proteome. There are some insights here and there about absence of structural homology, but no systematized, global evaluation of "modellable" and "unmodellable" regions. Therefore, I invite the authors to properly report on the complete structural coverage by answering, for example, the following questions:'*

- As indicated below, we have addressed the five specific questions raised by the reviewer. In addition, we have added further characterization of structural coverage by quantifying the total fraction of the proteome involved in mimicry, hijacking, and viral protein interactions, as well as the fraction mapped to known CATH families. We agree that including these statistics and analysis has improved the manuscript.

1a. *'How much of the total proteome is covered by the 872 models, and which proteins are the most/less studied?'*

- In the revised version of the manuscript, the total fraction of the viral proteome with matching structures (69%) is noted explicitly in the Abstract and in the opening paragraph of the Results.
- The fraction of modellable and unmodellable proteome covered is also visualized in Figure 1.
- We have also added Dataset EV8, which provides a detailed breakdown of the regions with structural coverage.
- We have added Dataset EV7, which presents a list of viral proteins sorted to show the best to least studied proteins.
- In addition, the three best studied proteins (NSP3, spike glycoprotein, NSP5) are now explicitly noted as such in their corresponding subsections in the Results.
- Also, the seven least studied 'dark' proteins (NSP2, NSP6, matrix glycoprotein, ORF6, ORF7b, and ORF9c, and ORF10) are explicitly noted as such, both in the Results and in the Discussion section on Suspects.
- Finally, we have added a new 'Structural Coverage' section to the Discussion that outlines the significance of the above results.

1b. *'What is the distribution of homology across the total proteome?'*

- Figure 1 shows the number of structural matches found for each viral protein.
- We have added Dataset EV7, which explicitly lists the number of structural matches for each viral protein.

1c. 'The regions that do not have any homology yet: Is it because of high disordered content or another factor?'

- This question is now explicitly addressed in the new 'Structural Coverage' section to the Discussion, with further details provided in Dataset EV8. Only ~4% of the dark proteome was predicted to be disordered, compared with ~2% for the non-dark proteome. Thus, most (96%) of the dark proteome is not accounted for by disorder, and remains largely unexplained, consistent with our previous observations (Perdigão et al, 2015).

1d. 'What is the overall content in secondary structure in this proteome as compared to other (viral) proteomes?'

- This question is now explicitly addressed in the new 'Structural Coverage' section to the Discussion, and detailed in Dataset EV12. Our analysis found very little difference in total secondary structure content in SARS-CoV-2 compared to either SARS-CoV or MERS-CoV. Certainly this question could be explored in more depth, but we do not think such an exploration would be warranted, given the intended audience of our work.

1e. 'Are the folds of SARS-CoV-2 proteins redundant (e.g. found more than 1 time among SARS-CoV-2 proteins) or rare as compared to other (viral) proteomes?'

- This question is now explicitly addressed in the new 'Structural Coverage' section to the Discussion, and detailed in Dataset EV9 (see 'Recurrence' column). Several folds are indeed redundant within the SARS-CoV-2 proteome, indicating these fulfill key viral functions. However, the exact same folds occur in SARS-CoV, at least according to assignments derived using the current CATH database.

2. 'The database has to be automatically updateable. Due to the COVID-19 pandemic, more structures are expected to be deposited in the PDB (or even modelled by other groups). Reported results, in a few months, will render the database obsolete, and of no use to researchers. Therefore, automatically updating the structural coverage of all proteins in a weekly manner and including links to other related resources will make the database immensely useful for the community and a solid website for researchers to find out the latest structural coverage updates for SARS-CoV-2 proteins.'

- We agree that, due to the fast moving nature of COVID research, automated inclusion of the latest PDB files is important. Thus we have modified our database update pipeline to enable newly published PDB files to be imported on a weekly basis, and to be immediately available via the Aquaria user interface. Achieving this has required two months of development combined with interactive, manual testing until we were sure that all SARS-CoV-2 structures to date can be automatically incorporated correctly.
- However, updating sequence-to-structure alignments to find all mappings to related PDB entries continues to require significant manual validation steps; in addition, each update also incurs significant AWS costs, so we plan to run them quarterly, as noted in the

manuscript. In principle, it would be possible to automate these steps, but this is currently out of scope as it would require securing funding beyond what we currently have. In any case, we believe we have addressed the key concern for most of our targeted readership, which will be to access new structures determined for SARS-CoV-2 proteins.

- We also note that the structural coverage map (Figure 1) is not automatically updated. In fact, each update to this graphic has taken two of us 2-3 months of full-time, mostly manual work to analyse all related structural models (now over 2,000). We consider this work necessary to reach the quality level we consider acceptable for publication.
- This graphic has many subtle elements that would be extremely difficult to fully automate; achieving this would be a multi-year project, so is out of scope. However, we are making progress towards automating a greatly simplified version that would still be useful for navigation. Given the very limited resources currently available in our team, we expect this will likely take 6-12 months of further development before this reaches a quality level we would consider to be ready for production.
- The reviewer suggested that we should include links to other related resources; it is unclear exactly what is intended here. Aquaria already links to UniProt and PDB. In addition, using the new Features described in this work provides direct links to CATH, SNAP2, and PredictProtein. Possibly the reviewer intended that we add more links to other COVID resources? If so, we would push back on this to some extent. We would make the general point that just adding multiple links to other resources can be counterproductive, potentially causing confusion rather than helping users. This may be especially true for the rapidly changing COVID resources. To avoid this confusion, each added link or integration with other resources needs care in planning, design, and execution. We would also argue that the effective integration of COVID resources is the role of the several prominent organizations that have received funding for such work. Currently, our core team has no such funding; however, we are certainly keen to work with these teams and we plan to do so once our manuscript is finally accepted.

3. 'The database has to be publicly shared and researchers have to be able to completely download its components. The most straightforward way that this can be achieved is by providing the database for download and installation on researchers' own host machines.'

- The complete Aquaria database has been publicly available since 2015. As indicated in the Reagents and Tools table, the latest version of the full database is available on Zenodo at <https://doi.org/10.5281/zenodo.4279163>.
- As indicated in the Data Availability section, a much smaller portion of this database containing only the most recent SARS-CoV-2 sequence-to-structure alignments is available at <https://doi.org/10.5281/zenodo.4934860>.
- In addition, the revised version of the manuscript now includes 12 supplementary datasets, covering all aspects of the data presented in this work.

4. *'Novelty aspect of the manuscript has to be re-evaluated: Here, the authors have included two rather daring aspects which I highly suggest toning down.'*

4a. *'The title: There is no evidence in a computational homology-based work for disruption of host immunity. Computational work is critical, but it is based on data collection. This has been very nicely done in this manuscript. In addition, it is based on analysis, which has also been performed accurately in this work. However, claims of relationships between biological phenotypes and data-mined models have to be supported by experimental data. Therefore, I invite the authors to change their title pointing to the strengths of their study, those being the (a) structural coverage map, (b) the visualization and open-source nature of their project, (c) the updateability of their data, as I suggested in (1), and (d) the direct homology-based comparisons that provide hypotheses on protein function but not on cellular effects, which are much more complex.'*

- The title has been revised, as suggested, to tone down claims related to immunity, and highlight instead the more specific and strongly supported outcomes from this work.
- We have removed the more speculative paragraphs discussing how our outcomes relate to host immunity and disease progression. We still think these are interesting speculations, but we now plan to re-purpose them in subsequent, more focused publications.
- In the revised manuscript, all remaining speculations are explicitly labelled as such.

4b. *'There is no modeling of protein-protein interactions for the viral proteins anywhere in this paper. This is not necessarily bad; the authors focus on single proteins and it is fine. However, if the authors do not perform this, then all claims in the manuscript regarding complexes have to be down toned or removed, because these come from authors' individual observations and not from a systematized study of protein-protein interactions of viral proteins. This includes, but is not limited to, Fig. 4B, which I suggest to completely remove: Structure is known, modeling has been done by several groups already (also MD), glycans are missing, conformation changes are not discussed, insights into binding intermediates are not presented, loop of RBD domain is not systematically described and how residues affect the interaction etc. Just showing a Figure panel of the complex is out of the scope of this paper, to my opinion.'*

- We agree that detailed, systematic modelling of protein-protein interactions would require additional tools and methods that are beyond the scope of our present study. In the revised manuscript, the first paragraph of the Discussion now mentions this and references some of the methods that would be required.
- Nonetheless, our study does systematically generate a list of ~30 putative protein-protein interactions, based on co-occurrences in the same matching structures

(Dataset EV4). Most of these interactions are directly supported by at least one structure determined using SARS-CoV-2 proteins. In these cases, we are simply noting and citing interaction claims made by other researchers. Thus, to our opinion, removing all discussion on interactions would be a disservice to our intended readership.

- We have therefore opted for the reviewer's alternative suggestion of toning down novel claims about interactions. Each inferred interaction based on structure of homologous proteins has been manually assessed, both by visually examining relevant structures, and by reading source literature. Each such manual assessment is now described in the Results section of the revised manuscript, and the assessment outcomes are explicitly stated in Datasets EV4-EV6.
- As a further response to this reviewer comment, we decided that it would be best to remove any interactions with marginal or insufficient evidence from the figures in the main text (Figs 1-4). In addition, we have added a figure to the Appendix (Fig. S1) focused on illustrating two of these interactions, as an illustration of how our Aquaria resource can be used to assess when evidence for an interaction is insufficient.
- Furthermore, as suggested by the reviewer we have completely removed the image previously shown in Fig. 4B; we also removed the previous Fig. 4A, which showed a somewhat similar analysis for NSP3. In the revised manuscript, we have replaced these images with Fig. 3, which has a much simpler goal, aimed at providing a visual summary of all structural evidence of hijacking - with the key take-away being that very little of the proteome is implicated. This new figure also partly addresses the first suggestion of reviewer #1, to better characterise proteome-wide structural coverage. Finally, together with Fig. S1, Fig. 3 also illustrates the utility of Aquaria in helping users generate hypotheses, then test them by finding and assessing specific structure-based evidence about protein-protein interactions.

REVIEWER #2

1. 'How do the authors discriminate mimicry from hijacking? What I know is that protein mimicry is a molecular feature that drives hijacking. It seems to me that they consider a viral protein to mimic a host protein, but not hijack a process, if none of the structures mapping to the viral protein include an interaction partner from the host? If that is true, I think it would be better to make this definition clear at the beginning of the Discussion before they start talking about the mimicry and hijacking groups, where they leave it for the readers to conclude on their own.'

- Our intention is to use the terms viral mimicry and hijacking in the normally accepted sense, thus in the Introduction we cite well-known publications to establish these terms (Elde & Malk, 2009; Davey et al. 2011). To clarify: we do not consider that viral hijacking **requires** structural evidence - it seems that our previous manuscript was unclear on this point. The confusion may have arisen because, within the coverage map (Fig. 1) and in

the Discussion section on Hijacking, our goal is to draw attention to the few cases where direct structural evidence of hijacking is available. However, we do not mean to suggest that lack of evidence implies that hijacking does not occur

- To hopefully avoid this misunderstanding, the revised Discussion section on Hijacking now begins with a restatement of the definition of hijacking and of the corresponding Davey et al. citation. The revised text in this paragraph hopefully more clearly explains that the cases highlighted are those that have structural evidence, but that additional processes are likely to be hijacked.
- Similarly, the revised section on Mimics also begins with a clear restatement of the definition of mimicry, and the corresponding Elde & Malk citation. The section also explicitly discusses several hijacked processes for which there is no direct structural evidence.

2. 'The authors state on Page 9 regarding the rare cases they found for viral mimicry or hijacking that "This may indicate that host interactions are rarely used in COVID-19 infection, consistent with the notion that viral activity is largely shielded from the host. However, other experimental techniques have found many more interactions between viral proteins (Pan et al., 2008), and with host proteins (Gordon et al., 2020). Thus, the small number of interactions found in this work likely indicates limitations in current structural data." This brings to my attention a couple of computational studies that have shown evidence of viral mimicry and hijacking at the larger interactome scale, using sequence/structural similarity approaches at the domain level as well as the residue level between viral proteins and human proteins. For example, this prior study: <https://www.pnas.org/content/108/26/10538>. While it is plausible that the rare cases of mimicry or hijacking found here may be either unique to this virus, or due to limitations in current structural data, an alternative hypothesis that needs to be ruled out is the specific criteria used here by the authors to identify homologous structures. How robust are the conclusions with regard to variations in these criteria? I recommend that the authors at least address this point in their discussion.'

- We fully agree with the general point that mining the available structure evidence using different methods and alternative criteria is likely to yield a very different number of interactions, hijackings, or mimicries. On the other hand, our modelling method (HHblits) is one of the most straightforward, is well established and widely used, and is likely to find all high-confidence models. Thus, I believe our core conclusions still stand: namely that currently available structural data are limited. In the revised manuscript, the first paragraph of the discussion now specifically references some of the additional methods that could be used to find additional interactions.

3. *'The notion of "feature sets" used throughout the paper may be confusing to some readers. As it first sounds like there are >32,000 unique features that are imported by the Aquaria server, but later in the methods it appears that these 32,000 features are more like values of a few (but significant) number of features.'*

- We agree that, in protein sequence analysis, the terms 'features' and 'feature sets' do not have universally understood meanings, and that this may make it difficult to clearly communicate how many features are available for SARS-CoV-2 sequences. Thus, in the revised manuscript, we have removed all mention of the specific number of features available for SARS-CoV-2 sequences. The opening paragraph of the Results now states that structures models "can be mapped with a wealth of features from UniProt, CATH (Dawson et al, 2017), SNAP2 (Hecht et al, 2015), and PredictProtein (Yachdav et al, 2014), in addition to user-defined features". Given that many predefined features are available, and that Aquaria can be enhanced with an unlimited number of user-defined features, we believe this statement to be accurate.

Thank you again for sending us your revised manuscript. We have now heard back from the two referees who were asked to evaluate the revised study. As you will see below, they are satisfied with the performed revisions and are supportive of publication.

Before we can formally accept the study for publication, I would ask you to address a few remaining editorial issues listed below.

REFEREE REPORTS

Reviewer #1:

The updated, revised manuscript of the authors reads really nice, it is of very high technical standards and will comprise a benchmark study for homology-based identification of coronaviruses with implications to the general fields of computational biology, structural biology and virology. I am completely satisfied with the authors' revisions and I congratulate and thank them for their critical work that I wholeheartedly recommend for immediate publication.

Reviewer #2:

All comments have been adequately addressed.

The authors have made all requested editorial changes.

Thank you again for sending us your revised manuscript. We are now satisfied with the modifications made and I am pleased to inform you that your paper has been accepted for publication.

Corresponding Author Name: Sean O'Donoghue

Manuscript Number: MSB-2020-10079